# Temporally Sparse Attack against Large Language Models in Time Series Forecasting

## Abstract

Large Language Models (LLMs) have recently demonstrated strong potential in zero-shot time series forecasting by leveraging their ability to capture complex temporal patterns through the next-token prediction mechanism. However, recent studies indicate that LLM-based forecasters are highly sensitive to small input perturbations. Existing attack methods, though, typically require modifying the entire time series, which is impractical in real-world scenarios. To address this limitation, we propose a Temporally Sparse Attack (TSA) against LLM-based time series forecasting. We formulate the attack as a Cardinality-Constrained Optimization Problem (CCOP) and introduce a Subspace Pursuit (SP)–based algorithm that restricts perturbations to a limited subset of time steps, enabling efficient and effective attacks. Extensive experiments on state-of-the-art LLM-based forecasters, including LLMTime (GPT-3.5, GPT-4, LLaMa, and Mistral), TimeGPT, and TimeLLM, across six diverse datasets, demonstrate that perturbing as little as 10% of the input can substantially degrade forecasting accuracy. These results highlight a critical vulnerability of current LLM-based forecasters to low-dimensional adversarial attacks.

## 1 Introduction

Time series forecasting is a critical tool across various domains, such as finance, traffic, energy management, and climate science. Accurate predictions of temporal patterns enable stakeholders to make informed decisions, optimize resources, and mitigate risks, thus playing a pivotal role in modern decision-making (Lim & Zohren, 2021; Liu et al., 2022b; Wang et al., 2024a).

Recently, Large Language Models (LLMs), originally designed for Natural Language Processing (NLP), have shown significant promise in capturing complex temporal dependencies across diverse scenarios (Garza & Mergenthaler-Canseco, 2023; Jin et al., 2024; Gruver et al., 2024). LLMs offer advanced capabilities, such as zero-shot forecasting, that allow them to generalize across various tasks without extensive retraining (Rasul et al., 2023; Ye et al., 2024; Liang et al., 2024). This positions LLMs as strong candidates for foundational models in time series forecasting.

Despite these strengths, LLMs are known to be susceptible to adversarial attacks, raising concerns about their reliability in critical applications (Zou et al., 2023; Liu et al., 2024). While LLM-based forecasters have demonstrated impressive accuracy (Ansari et al., 2024; Jiang et al., 2024), it remains uncertain whether decision-making processes can depend on these predictions in adversarial scenarios. Investigating the robustness of LLM-based models is therefore essential for ensuring their trustworthiness in real-world applications.

While adversarial attacks on machine learning models have been widely studied in computer vision and natural language processing domains (Wei et al., 2018; Xu et al., 2020; Morris et al., 2020), attacking LLMs in time series forecasting presents unique challenges. First, ground truth values (i.e., future time steps) cannot be used in attacks to prevent information leakage. Second, accessing the internal parameters, structure, and training data of LLMs is often infeasible for attackers, requiring attacks to operate under strict black-box conditions. Recent studies have explored gradient-free optimization techniques for adversarial attacks against LLM-based time series forecasters (Liu et al., 2025), demonstrating the feasibility of degrading model performance by perturbing the entire input series. However, such approaches present significant limitations in terms of practicality and imperceptibility. In real-world applications, particularly those involving time-sensitive data streams,

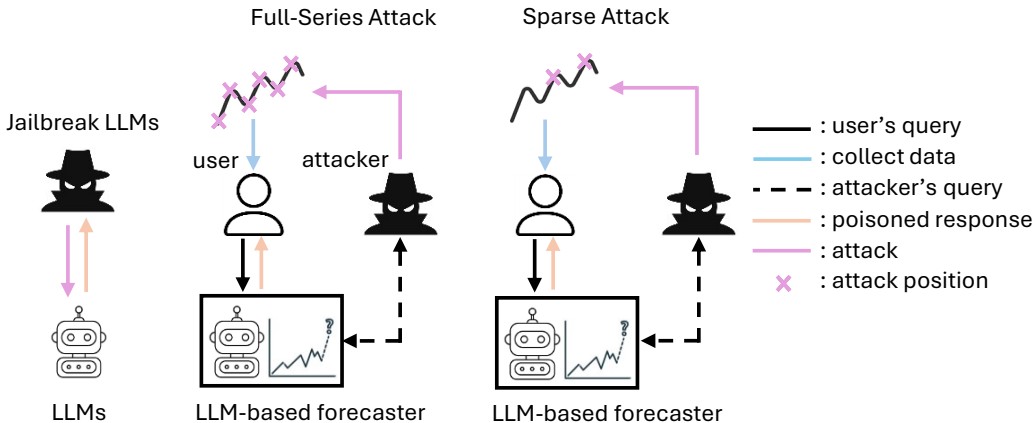

Figure 1: Conceptual comparison of three scenarios: (i) jailbreaking LLMs, (ii) full-series attacks on LLM-based forecasters, and (iii) the proposed TSA, which perturbs only a limited number of time steps. In adversarial time series settings, the threat model involves three key roles: the attacker, the user, and the forecaster. The key **gap** is that existing attacks require poisoning the entire input series.

the requirement to manipulate an entire time series, potentially spanning several hours, renders the attack infeasible and easily detectable. This concern motivates a more realistic and operationally relevant research question: **Can LLM-based forecasters be effectively disrupted by modifying only a small subset of the input time series?**

We address this question by proposing a Temporally Sparse Attack (TSA) framework (Figure 1) designed for highly constrained settings, where the adversary is limited to perturbing only a sparse subset of the input time series. This restriction aligns with realistic scenarios where imperceptibility and limited access are essential. To model the attack process, we formulate it as a Cardinality-Constrained Optimization Problem (CCOP) (Bhattacharya, 2009; Ruiz-Torrubiano et al., 2010). CCOP is inherently non-convex and NP-hard, and in this case, its resolution becomes even more challenging under black-box, label-free assumptions. To overcome these challenges, we adapt the Subspace Pursuit (SP) algorithm, originally developed for solving cardinality-constrained white-box LASSO problems (Dai & Milenkovic, 2009; Wang et al., 2012), to this black-box, label-free context by incorporating a gradient-free optimization strategy based solely on black-box queries to the forecasting model. TSA can effectively generate temporally sparse perturbations without access to ground truth labels or model internals, thereby offering a practical and stealthy solution suitable for real-world time series forecasting applications.

Our evaluation spans six LLM-based time series forecasting models across six diverse real-world datasets. The results demonstrate that TSA, which perturbs only 10% of the input data with small modifications, can still induce a substantial degradation in forecasting accuracy. Even filter-based defense mechanisms are largely ineffective against these attacks due to their sparse structure. These experiments empirically confirm that the proposed TSA is not only more stealthy but also more effective at bypassing filter-based adversarial defenses than full-series attacks. Overall, the findings highlight the urgent need to address such vulnerabilities in LLM-based forecasters to ensure their reliability in high-stakes applications.

## 2 RELATED WORK

**Sparse attacks in computer vision** aim to mislead recognition or detection models by perturbing only a small portion of the input image (Croce & Hein, 2019). The one-pixel attack (Su et al., 2019) employs a genetic algorithm to deceive deep neural network (DNN)-based image classifiers by modifying a single pixel, while GreedyFool (Dong et al., 2020) adopts a combination of greedy search and Projected Gradient Descent (PGD) to manipulate selected pixels in static settings. However, existing sparse attack studies predominantly operate under white-box assumptions, and the true label is typically required during perturbation generation. These assumptions do not hold in LLM-based

time series forecasting, which is inherently black-box and does not provide ground-truth labels during inference.

**Adversarial attacks on LLMs** have garnered significant attention, revealing how minor input manipulations can lead to substantial output alterations. These attacks are generally categorized into methods such as jailbreak prompting, where crafted prompts bypass safety guardrails to elicit unintended or harmful responses (Wei et al., 2024); prompt injection, embedding adversarial instructions within benign prompts to manipulate outputs (Greshake et al., 2023; Xue et al., 2024; Shen et al., 2024); gradient-based attacks, which exploit internal model parameters to create minimally invasive input perturbations (Zou et al., 2023; Jia et al., 2024); and embedding perturbations, which subtly alter input embeddings to disrupt the model's internal representations (Schwinn et al., 2024). While much of this research has focused on text-based tasks, the robustness of LLMs in non-textual domains like time series forecasting remains underexplored.

**Adversarial attacks in time series forecasting** have emerged as a critical research focus, exposing the vulnerabilities of forecasting models. Unlike static domains such as time series classification (Karim et al., 2020; Ding et al., 2023), time series forecasting presents unique challenges for adversarial research. One key constraint is the inability to use future ground truth values when generating perturbations, as this could lead to information leakage (Dang-Nhu et al., 2020; Liu et al., 2023). For example, in hourly temperature forecasting, the true label for 10 a.m. corresponds to the temperature at 11 a.m., which is unavailable to the user, the forecaster, and the attacker. To address this, surrogate labels have been introduced (Zhu et al., 2023; Lin et al., 2024), enabling attackers to bypass the need for ground truth. Most prior studies have concentrated on white-box scenarios (Xu et al., 2021; Liu et al., 2022a), where adversaries have full access to model parameters, structure, and training data. However, evaluating the robustness of LLM-based forecasting models presents additional complexities. These models typically operate in black-box settings, limiting access to their internal workings. Gradient-free black-box attacks have been proposed as a solution (Liu et al., 2025), but they often require modifying the entire time series, which is impractical.

## 3 THREAT MODEL

In what follows, we first provide an overview of LLM-based time series forecasting as the foundation of our study, and then formally define the goals and capability constraints of adversaries when conducting sparse attacks against LLM-based forecasters.

**LLM-Based Time Series Forecasting.** LLMs have shown great promise in time series forecasting by leveraging their next-token prediction capability. A typical LLM-based time series forecasting framework, denoted as $f(\cdot)$, comprises two key components: an embedding or tokenization module and a pre-trained LLM. The embedding module encodes time series into a sequence of tokens suitable for processing by the LLM, while the LLM captures temporal dependencies and autoregressively predicts subsequent tokens based on its learned representations.

Let $\mathbf{x}_t \in \mathbb{R}^d$ represent a $d$-dimensional time series at time $t$. Define $\mathbf{X}_t = \{\mathbf{x}_{t-T+1}, \ldots, \mathbf{x}_t\}$ as the sequence of $T$ recent historical observations and $\mathbf{Y}_t = \{\mathbf{y}_{t+1}, \ldots, \mathbf{y}_{t+L}\}$ as the true future values for the next $L$ time steps. The forecasting model $f(\cdot)$ predicts the future values from the historical observations, which is formulated as:

$$\hat{\mathbf{Y}}_t = f(\mathbf{X}_t), \tag{1}$$

where $\hat{\mathbf{Y}}_t$ denotes the predicted future values. Typically, the prediction horizon $L$ is constrained to be less than or equal to the historical horizon $T$, i.e., $L \leq T$. This ensures that the model leverages sufficient historical context while maintaining computational efficiency.

By effectively combining the embedding module's ability to encode raw time series data and the LLM's capacity to model complex temporal patterns, these models have become powerful tools for addressing a wide range of zero-shot forecasting challenges across various domains.

**Temporally Sparse Attack against LLM-based Forecaster.** The goal of attacking an LLM-based time series forecasting model $f(\cdot)$ is to manipulate it into producing abnormal outputs that differ substantially from their typical predictions and the actual ground truth, using minimal and nearly undetectable perturbations.

The adversarial attack can be modeled as a maximum optimization problem:

$$\max_{\boldsymbol{\rho}} \mathcal{L}\left(f\left(\mathbf{X}_t + \boldsymbol{\rho}\right), \mathbf{Y}_t\right) \quad \text{s.t.} \ \|\rho_i\|_p \leq \epsilon, i \in [t - T + 1, t], \tag{2}$$

where $\boldsymbol{\rho} = \{\rho_{t-T+1}, \ldots, \rho_t\}$ denotes the perturbations added into the clean historical time series $\mathbf{X}_t = \{\mathbf{x}_{t-T+1}, \ldots, \mathbf{x}_t\}$. Here, the loss function $\mathcal{L}$ measures the discrepancy between the model's predictions and the ground truth, while $\epsilon$ serves as a constraint on the perturbation magnitude under the $\ell_p$-norm, ensuring that the adversarial attack remains subtle and imperceptible. Typically, the global average $\bar{\mathbf{X}}$ serves as the reference point to determine whether the added perturbations are imperceptible. Consequently, $\epsilon$ is defined as a proportion of the global average, e.g., $\epsilon = 5\% \times \bar{\mathbf{X}}$.

The true future values $\mathbf{Y}_t$ are generally unavailable during the practical forecasting process. As a result, to avoid future information leakage, the ground truth $\mathbf{Y}_t$ is substituted with the predicted values $\hat{\mathbf{Y}}_t$ produced by the forecasting model. Specifically, in Equation 2, $\mathbf{Y}_t$ is replaced with $\hat{\mathbf{Y}}_t$. In practical applications, it is generally infeasible to access the complete set of detailed parameters of an LLM, compelling the attacker to approach the target model as a black-box system. In other words, no internal information of $f(\cdot)$ in Equation 2 is available.

The computed perturbations $\boldsymbol{\rho} = \{\rho_{t-T+1}, \ldots, \rho_t\}$ are typically applied across the entire input window, which makes full-series poisoning burdensome in practice. For example, for a 5-minute-ahead traffic forecaster that uses $T = 48$ input steps, an attacker would need to manipulate $48$ consecutive measurements, i.e., $48 \times 5 \ \text{minutes} = 4 \ \text{hours}$ of data. This example illustrates the practical difficulty of poisoning the entire series. In this study, we impose strict limitations on the attacker's capabilities, allowing them to pollute only $\tau$ time steps. Furthermore, since the future true values $\mathbf{Y}_t$ are unavailable, they are approximated using the predicted values $\hat{\mathbf{Y}}_t = f(\mathbf{X}_t)$. Under this constraint, the attack process is reformulated as a CCOP (Bhattacharya, 2009):

$$\max_{\boldsymbol{w}} \mathcal{L}\left(f\left(\mathbf{X}_t\left(1 + \boldsymbol{w}\right)\right), \hat{\mathbf{Y}}_t\right)$$
$$\text{s.t.} \ \|\boldsymbol{w}\|_0 = \tau \quad \text{and} \quad \|w_i\|_1 \leq \epsilon, \ i \in [t - T + 1, t], \tag{3}$$

where $\boldsymbol{w} = \{w_{t-T+1}, \ldots, w_t\}$ represents multiplicative adversarial perturbations. The cardinality constraint, also called $\tau$-sparse $\ell_0$-norm constraint, restricts the number of non-zero elements in adversarial perturbations to a fixed small number, ensuring that the adversarial perturbations are sparse on the temporal dimension. Besides, the $\ell_1$-norm constraint limits the magnitude of each non-zero perturbation, ensuring the modifications remain imperceptible.

It should be noted that the global average is unsuitable as a reference for the average magnitude of the manipulated series under the temporally sparse setting. Instead, each manipulated time step requires a unique reference point to ensure the magnitude of the perturbation at each time step is bounded. The limitation of the poisoned value at time step $i$ can be expressed as:

$$\|\mathbf{x}_i + \rho_i\|_1 = \|\mathbf{x}_i\left(1 + w_i\right)\|_1 \leq \|\mathbf{x}_i\left(1 + \epsilon\right)\|_1, \tag{4}$$

where $\|\rho_i\|_1 = \|w_i \cdot \mathbf{x}_i\|_1 \leq \|\epsilon \cdot \mathbf{x}_i\|_1$. Consequently, the additive perturbation $\mathbf{X}_t + \boldsymbol{\rho}$ in Equation 2 is replaced with the multiplicative perturbation $\mathbf{X}_t\left(1 + \boldsymbol{w}\right)$ in Equation 3.

Furthermore, in many real-world applications, adversaries often lack access to the complete training dataset, rendering it infeasible to exploit the data distribution or model training process directly. Given the preceding discussion, the **capabilities and constraints** of the attacker under the temporally sparse attack setting can be summarized as follows: **(i)** no access to the training dataset; **(ii)** no access to the internal architecture or parameters of the LLM-based forecasting model; **(iii)** no access to the ground truth values during inference; **(iv)** the ability to perturb only a sparse subset of the input time series; and **(v)** the capability to query the forecasting model in a black-box manner.

## 4 PERTURBATION COMPUTATION

The temporally sparse attack process is formulated as a CCOP in Equation 3, which is inherently non-convex and NP-hard. Subspace Pursuit (SP) has been shown to provide approximate solutions to cardinality-constrained white-box LASSO problems within polynomial time (Dai & Milenkovic, 2009; Wang et al., 2012). However, applying SP in the context of adversarial attacks against LLM-based forecasting introduces two major challenges: the unavailability of model parameters and the absence of ground truth labels. To overcome these constraints, we integrate gradient-free optimization techniques and adapt the SP algorithm.

## 4.1 $\tau$-SPARSE PERTURBATION COMPUTATION

To solve the optimization problem in Equation 3, we propose an adapted SP method, outlined as Algorithm 1. In our adaptation, the $\ell_1$-norm constraint is incorporated as a subroutine to maintain the imperceptibility of the perturbations. Here, the support set $S = \text{supp}(\boldsymbol{w}) = \{i : w_i \neq 0\}$ denotes the indices of nonzero elements in the perturbation vector $\boldsymbol{w}$, with $|S|$ representing its cardinality. To efficiently update the support set, we define the merge operator:

$$\mathcal{M}\left(\boldsymbol{w}_S, w_j\right) = \begin{cases} \boldsymbol{w}_S, & j \in S, \\ \{\boldsymbol{w}_S, w_j\}, & j \notin S. \end{cases} \quad (5)$$

This operator ensures that when a new candidate perturbation $w_j$ is selected, it is either retained in the existing support set $S$ if it is already present, or added as a new element if it is not.

Algorithm 1 describes the iterative process for estimating the sparse multiplicative adversarial perturbations $\boldsymbol{w}$. At each iteration, the algorithm identifies the indices corresponding to the $\tau$ largest loss values resulting from applying candidate perturbations. The candidate perturbations $w_j$ are computed using the gradient-free optimization technique as in Section 4.2. Then, the support set $S$ is updated by including the identified indices. The support set $S$ is subsequently refined by selecting the $\tau$ elements with the largest individual prediction loss. Any components outside the updated support set are reset to zero. This process repeats until the loss $\boldsymbol{r}$ converges and the final $\tau$-sparse multiplicative adversarial perturbation $\boldsymbol{w}$ is returned.

This method effectively enforces the CCOP by ensuring that only $\tau$ time steps are modified while maintaining a bounded perturbation magnitude. The adapted SP approach enables efficient selection of perturbation locations, ensuring maximal adversarial impact while keeping modifications imperceptible. Moreover, the computation complexity of the proposed method is $\mathcal{O}(T \times \tau)$, whereas a standard greedy algorithm has a significantly higher complexity of $\mathcal{O}(T^\tau)$.

1: **Input:** Time series $\mathbf{X} \in \mathbb{R}^{d \times T}$, the loss function $\mathcal{L}$, the LLM-based forecaster $f(\cdot)$, and sparsity level $\tau$ of the multiplicative adversarial perturbations $\boldsymbol{w}$.

2: **Initialize** the perturbation vector $\boldsymbol{w} := \boldsymbol{0}$ as zeros, the support set $S := \emptyset$ as an empty set, and the loss value $\boldsymbol{r} := 0$ as zero.

3: **Return** $\boldsymbol{w} = \{w_t\}$ for $t \in [0, \ldots, T-1]$.

4: **while** not converged **do**

5:     Find $\ell$ as the index set of the $\tau$ largest losses of $f(\mathbf{X}_t(1 + \mathcal{M}(\boldsymbol{w}_S, w_j)))$ in which $w_j$ is the candidate perturbation, where $j \in [1, \ldots, T] \,\&\, j \notin S$.

6:     Update the support set $S := S \cup \{\ell\}$.

7:     Update the sparse vector $\boldsymbol{w}_S := \epsilon \cdot \text{sign}(\hat{\boldsymbol{g}}_S)$.

8:     Update the support set $S$ as the index set of the $\tau$ largest losses of $f(\mathcal{X}_t(1 + w_i))$ for $i \in S$.

9:     Set $w_i = 0$ for all $i \notin S$.

10:     Update $\boldsymbol{r} := \mathcal{L}\left(f(\mathcal{X}_t(1 + \boldsymbol{w}_S)), \hat{\mathbf{Y}}_t\right)$.

11: **end while**

12: **Return** the $\tau$-sparse multiplicative adversarial perturbations $\boldsymbol{w}$.

Algorithm 1: Computing $\boldsymbol{w}$ with adapted SP.

## 4.2 CANDIDATE PERTURBATION

The candidate perturbation in the first step of Algorithm 1 (line 5) is to perturb the specific time step $j$, which can be formulated as:

$$\max_{w_j} \mathcal{L}\left(f\left(\mathbf{X}_t + \{0, \ldots, w_j \cdot \mathbf{x}_i, \ldots, 0\}\right), \hat{\mathbf{Y}}_t\right) \quad \text{s.t.} \ \|w_j\|_1 \leq \epsilon. \quad (6)$$

Here, the perturbation $w_j$ is applied only at time step $j$. The magnitude of the perturbation is bounded by the constraint $\epsilon$, while maximizing the impact on the loss function $\mathcal{L}$.

In the black-box setting, Equation 6 cannot be solved using gradient-based methods such as Stochastic Gradient Descent (SGD). Instead, a gradient-free optimization technique can be employed to estimate the gradients, as follows:

$$\hat{g} = \frac{\mathcal{F}(\mathbf{X}_t, w_j, \Delta) - \mathcal{F}(\mathbf{X}_t, w_j, -\Delta)}{2 \cdot \Delta}, \quad (7)$$

where $\hat{g}$ represents the estimated gradients, $\Delta$ denotes a random Gaussian noise, and $\mathcal{F}(\mathbf{X}_t, w_j, a) = f(\mathbf{X}_t + \{0, \ldots, (w_j + a) \cdot \mathbf{x}_i, \ldots, 0\})$ denotes querying the target model with a noise term $a$.

Similar to the Fast Gradient Sign Method (FGSM) (Goodfellow et al., 2015), the perturbation can be computed using the estimated gradients $\hat{g}$ as $w_j = \epsilon \cdot \text{sign}(\hat{g})$, where $\text{sign}(\cdot)$ denotes the signum

Table 1: Comparison of adversarial attacks. In this table, TS denotes time series.

| Method | Black-box | Label-free | No train set | Applicable to LLMs in TS | Temporal sparsity |
|---|---|---|---|---|---|
| TS forecasting attacks (Dang-Nhu et al., 2020) | ✗ | ✓ | ✓ | ✗ | ✗ |
| Black-box attacks (Guo et al., 2019) | ✓ | ✗ | ✗ | ✗ | ✗ |
| Attacks against LLMs in TS (Liu et al., 2025) | ✓ | ✓ | ✓ | ✓ | ✗ |
| Proposed temporally sparse attack (TSA) | ✓ | ✓ | ✓ | ✓ | ✓ |

function. This approach ensures that the perturbation magnitude is bounded by $\epsilon$ while aligning with the direction of the estimated gradients.

After generating the candidate perturbation, the corresponding loss is computed as

$$r := \mathcal{L}\Big(f(\mathbf{X}_t(1 + \mathcal{M}(\boldsymbol{w}_S, w_j))), \hat{\mathbf{Y}}_t\Big), \tag{8}$$

which functions as the ranking index in the initial step of the iterative procedure in Algorithm 1.

**Brief Discussion.** Adversarial attacks against LLMs for time series forecasting remain extremely limited, primarily due to the following practical constraints: **(i)** no access to true labels at inference time, in order to avoid future information leakage; **(ii)** no access to model parameters, as LLMs are prohibitively large and impractical for attackers to obtain; and **(iii)** no access to training data, since LLM-based forecasters operate in a zero-shot setting and are trained on massive, heterogeneous datasets drawn from diverse applications. TSA satisfies all these constraints while additionally operating in a temporally sparse setting, making it well-suited for realistic LLM-based forecasting scenarios. Table 1 presents a simple comparison between the proposed TSA and existing methods.

## 5 EXPERIMENT

In this section, we evaluate the effectiveness of TSA across six datasets and seven forecasting models, including six LLM-based and three non-LLM-based baselines, in comparison with two existing attacks. We primarily address the following potential concerns: **Q1.** Does the proposed TSA significantly degrade the predictive performance of LLM-based forecasters? **Q2.** What explains the effectiveness of TSA? **Q3.** Can TSA bypass existing adversarial mitigation strategies? **Q4.** How sensitive is TSA to different hyperparameter choices?

Detailed experimental settings are provided in Appendix B. In summary: **Baseline attacks.** For comparison, we consider Gaussian White Noise (GWN) and a full-series attack, Directional Gradient Approximation (DGA) (Liu et al., 2025). Besides, we construct two sparse variants of DGA, which perturb the same number of time steps as the proposed TSA, but select the attack positions either through random sampling or via a greedy search strategy. **Target models.** We evaluate TSA against state-of-the-art forecasting systems, including TimeGPT (Garza & Mergenthaler-Canseco, 2023), TimeLLM (Jin et al., 2024), LLMTime (Gruver et al., 2024) with GPT-3.5, GPT-4, LLaMa2, and Mistral as backbone models, as well as non-LLM forecasters including TimesNet (Wu et al., 2023), TimeMixer (Wang et al., 2024a), and TimeXer (Wang et al., 2024b). **Datasets.** Experiments are conducted on six real-world datasets, ETTh1, ETTh2, Traffic, Weather, Exchange, and Solar, spanning domains such as electricity, transportation, geoscience, economics, and energy. **Metrics.** Forecasting performance is evaluated using two standard error measures: Mean Absolute Error (MAE) and Mean Squared Error (MSE).

Additional experiments are reported in the Appendix. Specifically, the evaluations on long input/output horizons and variate-wise forecasting are provided in Appendix C and Appendix D, respectively. Appendix E empirically analyzes the trade-off between effectiveness and efficiency in single-query versus multi-query attacks. Appendix F compares the proposed TSA with two sparse variants of DGA, further demonstrating the strength of the SP-based solution. Appendix G presents a vulnerability comparison between LLM-based and non-LLM-based forecasters. Appendix H extends the proposed TSA to a targeted attack setting to evaluate whether an adversary can force the forecasting model to produce attacker-specified outputs. Appendix I empirically analyzes the computational cost of the SP-based solution and the greedy search strategy. Appendix J examines the reliability of the attack performance by running the proposed attack multiple times and reporting the variance.

Table 2: Adversarial attack effectiveness comparison. Forecasting models process each variate independently, treating the multivariate task as a collection of univariate forecasting problems. A fixed input length of 96 and an output length of 48 are used across all models and datasets. Lower MSE and MAE values indicate better predictive performance. For TSA and DGA, the perturbation magnitude constraint is fixed at $\epsilon = 0.1$, while for GWN, the deviation is set to 2% of the mean value of each dataset. For clarity, the worst and second-worst performance for each dataset–model combination are highlighted in bold and italics. The sparsity level of TSA is set to $\tau = 9$, while both DGA and GWN poison the entire input series.

| Models | LLMTime w/ GPT-3.5 | | LLMTime w/ GPT-4 | | LLMTime w/ LLaMa 2 | | LLMTime w/ Mistral | | TimeLLM w/ GPT-2 | | TimeGPT (2024) | | TimesNet (non-LLM) | |
|---|---|---|---|---|---|---|---|---|---|---|---|---|---|---|
| Metrics | MSE | MAE | MSE | MAE | MSE | MAE | MSE | MAE | MSE | MAE | MSE | MAE | MSE | MAE |
| Traffic | 0.837 | 0.844 | 0.805 | 0.779 | 0.891 | 1.005 | 0.826 | 0.973 | 0.995 | 1.013 | 1.890 | 1.201 | 1.095 | 1.022 |
| w/ GWN | 0.882 | 0.908 | 0.883 | 0.864 | 0.917 | 1.063 | 1.054 | 1.031 | 1.123 | 1.221 | 1.848 | 1.204 | 1.103 | 1.035 |
| w/ DGA | **0.955** | **1.073** | **1.417** | **1.214** | **0.994** | 1.083 | **1.744** | **1.217** | **1.161** | *1.328* | *1.918* | **1.218** | **1.155** | *1.047* |
| w/ TSA | *0.901* | *1.037* | *1.179* | *1.008* | *0.969* | **1.085** | *1.493* | *1.204* | *1.147* | **1.332** | **1.920** | *1.208* | *1.136* | **1.093** |
| ETTh1 | 0.073 | 0.213 | 0.071 | 0.202 | 0.086 | 0.244 | *0.097* | 0.274 | 0.089 | 0.202 | 0.059 | 0.192 | 0.073 | 0.202 |
| w/ GWN | 0.077 | 0.219 | 0.076 | 0.213 | 0.087 | 0.237 | 0.094 | 0.291 | **0.102** | 0.231 | 0.059 | 0.193 | 0.074 | 0.202 |
| w/ DGA | **0.085** | **0.249** | **0.083** | **0.232** | *0.091* | **0.251** | **0.098** | **0.295** | *0.099* | **0.248** | *0.060* | *0.198* | **0.081** | **0.213** |
| w/ TSA | *0.082* | *0.235* | *0.079* | *0.230* | **0.092** | *0.249* | 0.097 | **0.295** | 0.091 | *0.237* | **0.061** | **0.203** | *0.080* | *0.206* |
| ETTh2 | 0.263 | 0.372 | 0.155 | 0.267 | 0.237 | 0.373 | 0.277 | 0.492 | 0.238 | 0.361 | 0.161 | 0.297 | 0.166 | 0.316 |
| w/ GWN | 0.263 | 0.342 | 0.175 | 0.303 | 0.231 | *0.429* | 0.346 | 0.505 | 0.235 | 0.355 | 0.160 | 0.301 | 0.166 | 0.316 |
| w/ DGA | **0.275** | **0.408** | **0.201** | **0.327** | *0.257* | 0.425 | **0.356** | **0.554** | **0.302** | **0.441** | **0.171** | **0.312** | **0.169** | **0.321** |
| w/ TSA | *0.271* | *0.402* | *0.195* | *0.319* | **0.258** | **0.432** | *0.350* | *0.547* | *0.299* | *0.440* | *0.168* | *0.307* | *0.167* | *0.319* |
| Weather | 0.005 | 0.051 | 0.004 | 0.048 | 0.008 | 0.072 | 0.006 | 0.057 | *0.004* | 0.034 | 0.004 | 0.043 | 0.003 | 0.042 |
| w/ GWN | 0.005 | 0.053 | 0.005 | 0.051 | 0.008 | 0.074 | **0.007** | **0.066** | *0.004* | 0.033 | 0.004 | 0.043 | 0.003 | 0.042 |
| w/ DGA | **0.006** | **0.063** | **0.006** | **0.061** | 0.009 | **0.079** | 0.007 | 0.062 | **0.005** | **0.052** | 0.006 | *0.071* | **0.004** | **0.045** |
| w/ TSA | **0.006** | *0.060* | **0.006** | *0.058* | **0.010** | 0.076 | 0.006 | 0.065 | 0.004 | 0.048 | **0.007** | **0.072** | 0.004 | *0.043* |
| Exchange | 0.038 | 0.146 | 0.040 | 0.152 | 0.043 | 0.167 | 0.151 | 0.274 | 0.056 | 0.188 | 0.256 | 0.368 | 0.056 | 0.184 |
| w/ GWN | 0.042 | 0.179 | 0.046 | 0.182 | 0.050 | 0.185 | 0.160 | 0.298 | 0.059 | *0.194* | 0.329 | 0.413 | **0.065** | **0.195** |
| w/ DGA | **0.058** | **0.224** | **0.068** | **0.199** | **0.069** | **0.213** | **0.219** | **0.303** | **0.077** | **0.256** | **0.578** | **0.556** | *0.062* | *0.194* |
| w/ TSA | *0.049* | *0.196* | *0.065* | *0.190* | *0.059* | *0.210* | *0.190* | *0.299* | *0.061* | 0.189 | *0.474* | *0.537* | *0.062* | 0.190 |
| Solar | 0.316 | 0.325 | 0.235 | 0.276 | 0.297 | 0.304 | 0.303 | 0.314 | 0.331 | 0.347 | 0.244 | 0.279 | 0.301 | 0.319 |
| w/ GWN | 0.319 | 0.323 | 0.236 | 0.280 | 0.299 | 0.304 | 0.305 | 0.315 | *0.337* | 0.348 | 0.244 | 0.282 | 0.305 | 0.322 |
| w/ DGA | **0.342** | *0.355* | **0.291** | *0.308* | **0.315** | **0.318** | **0.327** | **0.346** | **0.340** | **0.354** | *0.281* | *0.306* | **0.317** | **0.330** |
| w/ TSA | **0.342** | **0.364** | *0.288* | **0.310** | *0.307* | *0.314* | *0.325* | *0.339* | *0.337* | *0.351* | **0.290** | **0.315** | *0.312* | *0.326* |

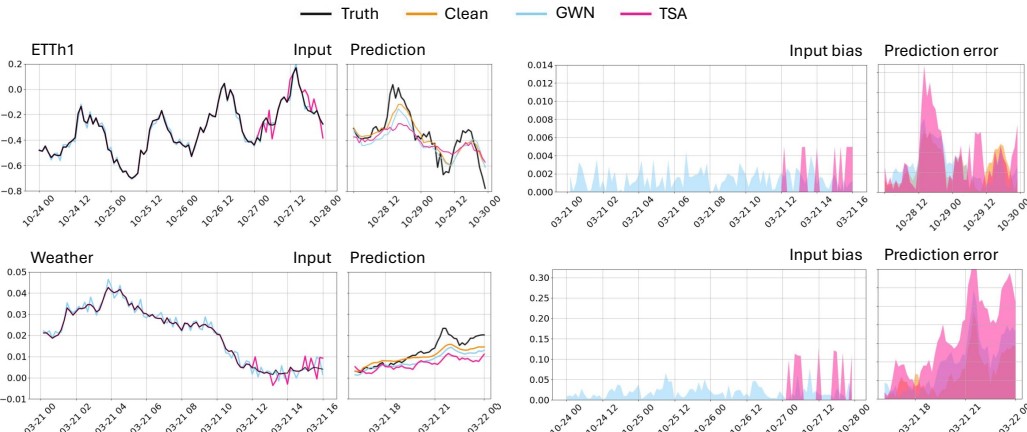

Figure 2: Comparison of input bias and prediction errors under different attack settings. *Top*: LLMTime with GPT-3.5 on the ETTh1 dataset. *Bottom*: TimeGPT on the Weather dataset.

## 5.1 EFFECTIVENESS ANALYSIS AND VISUALIZATION (**Q1**)

TSA induces up to a $4\times$ increase in prediction errors for LLM-based time series forecasters compared to GWN across a range of real-world applications. As shown in Table 2, TSA significantly increases both MSE and MAE across most models and datasets, demonstrating its strong impact on degrading LLM-based forecasting performance. The Traffic dataset shows the greatest deterioration, with

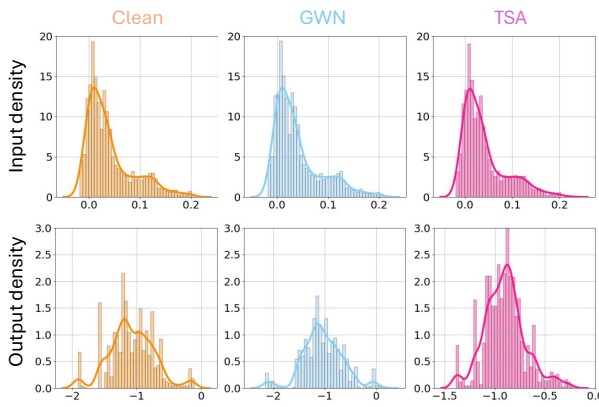 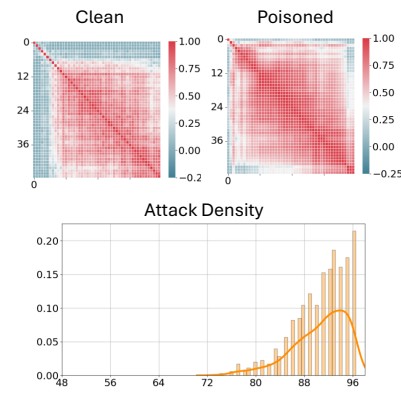

Figure 3: Input and output distributions for LLMTime with GPT-3.5 on ETTh1 under clean input, GWN, and the proposed TSA.

Figure 4: *Top*: Correlation matrices of prediction errors with and without the proposed TSA. *Bottom*: Attack position distribution.

TSA increasing errors by 80.75% for LLMTime w/ Mistral and 46.45% for LLMTime w/ GPT-4, highlighting the models' vulnerability.

Despite perturbing only 9 out of 96 time steps, TSA achieves degradation in forecasting performance that is largely comparable to the full-series attack DGA. For example, across datasets such as Traffic, ETTh2, and Solar, the MSE/MAE values under TSA are often close to or even match those obtained with DGA. This demonstrates that sparse perturbations can be just as disruptive as full-series modifications. In contrast to DGA, which requires modifying the entire input and repeated model queries, TSA reaches similar effectiveness with significantly fewer perturbations, underscoring its practicality in real-world adversarial scenarios. The results further confirm that incorporating CCOP and SP techniques effectively enhances the attack's precision. In Appendix F, we further compare the proposed TSA with two sparse variants of DGA. TSA produces approximately 84% larger prediction errors than the sparse DGA with greedy search and more than 127% larger errors than the sparse DGA with random position selection.

Figure 2 illustrates a direct comparison between GWN and TSA in terms of input perturbations and their effect on forecasting errors. For both LLMTime w/ GPT-3.5 on ETTh1 and TimeGPT on Weather, GWN introduces small, uniformly distributed fluctuations across the input series, while TSA injects sparse, localized perturbations into only 10% of the time steps. This effect is visible in the *right* panels, where TSA produces significantly larger prediction errors than GWN. Notably, TSA-induced errors align with critical regions of the time series (e.g., sharp rises or drops), demonstrating that the attack effectively exploits model vulnerabilities rather than merely injecting noise.

## 5.2 INTERPRETATION AND UNDERSTANDING (**Q2**)

Figure 3 compares input and output distributions under clean input, GWN, and TSA. While the input distributions show minor differences across all cases, the output distribution under TSA deviates significantly, indicating that TSA exerts a stronger adversarial effect than GWN by disrupting model forecasts more severely.

Figure 4 provides insights into the structural effects of TSA on prediction errors and its temporal attack distribution. The *top* panels compare the correlation matrices of prediction errors under clean and attacked settings. Under TSA, the correlation matrix exhibits stronger and more widespread correlations across time steps, revealing that sparse perturbations introduce structured distortions that propagate through the forecast horizon. This demonstrates that TSA does not merely inject noise but systematically alters the temporal dependencies leveraged by LLM-based forecasters.

Figure 4 *bottom* panel illustrates the distribution of attack positions on the ETTh1 dataset. The histogram shows that TSA tends to concentrate perturbations toward later portions of the input sequence, where the influence on the output forecast is strongest. This sparse-yet-targeted strategy

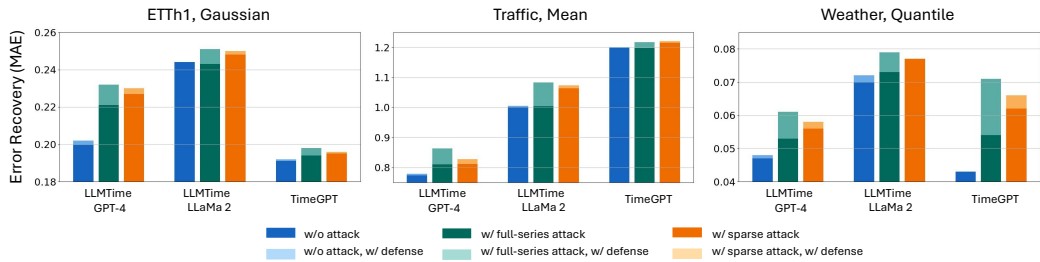

Figure 5: Full series and temporally sparse adversarial attacks on different LLM-based forecasting models protected by filter-based adversarial defense strategies. Light green and light orange indicate the recovered prediction error. The full series attack is DGA (Liu et al., 2025).

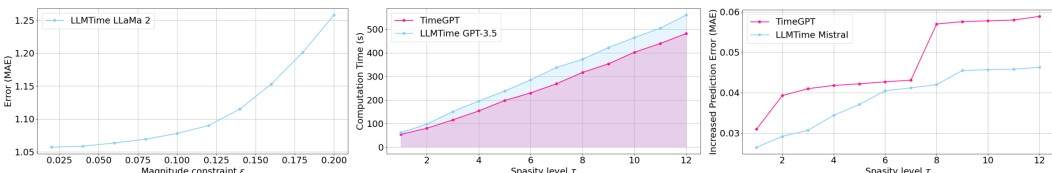

Figure 6: Hyperparameter sensitivity analysis. *Left* illustrates how the prediction errors increase exponentially as the perturbation magnitude constraint grows. *Middle* demonstrates that computational cost scales linearly with the sparsity level. *Right* shows that the prediction errors increase as the sparsity level of perturbations rises.

explains why TSA achieves significant adversarial impact with limited perturbations, reinforcing its efficiency and stealth compared to random noise injection.

## 5.3 MITIGATION BYPASSING TEST (Q3)

This section evaluates whether TSA can bypass adversarial defenses. DGA, a black-box attack against LLM-based forecasters (Liu et al., 2025) that perturbs the full input series, serves as a baseline. Three filter-based defenses, including Gaussian, Mean, and Quantile filters (Xie et al., 2019), are applied without requiring re-training or fine-tuning of the LLM-based forecasters.

Figure 5 shows that these defenses fail to recover errors under TSA (minimal light orange bars), but effectively mitigate full-series attacks (larger light green bars). This suggests that TSA's sparse, concentrated modifications are harder to correct than full-series attacks, which distribute perturbations more uniformly, allowing them to be smoothed by filtering techniques. By modifying only a limited number of steps, TSA bypasses the statistical assumptions on which many filtering defenses rely. Consequently, the sparse perturbations introduce structured errors that persist through the forecast horizon, leading to significant degradation in model performance despite the application of defenses.

## 5.4 HYPERPARAMETER SENSITIVITY ANALYSIS (Q4)

There are two key hyperparameters in Algorithm 1: the perturbation magnitude constraint $\epsilon$ and the sparsity level $\tau$. In this section, we analyze their impact on the effectiveness and computational cost of TSA, as illustrated in Figure 6.

The *left* panel demonstrates that as $\epsilon$ increases, the prediction errors of LLMTime with LLaMa 2 on Traffic grow exponentially. The magnitude constraint balances the imperceptibility and the attack effectiveness. The *middle* panel shows that the computational cost of TSA scales linearly with the sparsity level $\tau$, meaning that increasing the number of perturbed time steps results in a proportional rise in computation time. The *right* illustrates that the prediction errors of TimeGPT and LLMTime with Mistral increase as $\tau$ rises, though the impact varies across models, with TimeGPT showing a more pronounced error increase at higher sparsity levels. These results suggest a trade-off between attack efficiency and computational complexity.

## 6   POTENTIAL MITIGATION DISCUSSION

Finally, we discuss potential strategies to mitigate TSA and enhance the resilience of LLM-based forecasting. Although adversarial training (Zhang, 2018; Madry et al., 2018) is a common defense in deep learning, it is impractical here due to the high computational costs of fine-tuning LLMs. Additionally, as discussed in Section 5.3, filter-based defenses fail to counter TSA effectively, as TSA's sparsity can bypass the statistical assumptions underlying these defenses.

A simple but novel autocorrelation-based detection method may be effective, which leverages the zero-shot capability of LLM-based forecasting models. Specifically, the forecaster is used to backcast historical time series from its own predictions, which are then compared with the original inputs to identify manipulated time steps. Once detected, the reformation is applied to correct the poisoned inputs. This approach exploits the autocorrelation properties of time series to detect sparse adversarial modifications without requiring external training.

## 7   CONCLUSION

This work presents a Temporally Sparse Attack (TSA), designed for LLM-based time series forecasting models in constrained adversarial scenarios, where only a small subset of input time steps can be modified. We model the attack as a Cardinality-Constrained Optimization Problem (CCOP) and develop a Subspace Pursuit (SP)-based method to efficiently generate sparse perturbations. TSA operates in a black-box setting, requiring no access to future data or internal model parameters.

Experiments on advanced LLM-based time series forecasting models across diverse real-world datasets show that perturbing only a small portion of the input significantly degrades forecasting performance. LLM-based forecasters exhibit high sensitivity to adversarial manipulation. Our findings demonstrate that conventional filter-based approaches fail to mitigate TSA, emphasizing the importance of enhancing robustness in time series foundation models. This research provides a framework for improving the resilience of AI systems and supports future advancements in Trustworthy AI.

ETHICS STATEMENT

This research explores the robustness and vulnerability of large language models in time series forecasting, which has critical applications in domains such as transportation, finance, and healthcare. As these models become increasingly integral to real-world decision-making, understanding and mitigating their susceptibility to adversarial attacks is essential for the development of trustworthy and reliable AI systems.

Our work aims to enhance the resilience of time series models against adversarial threats by contributing insights into attack strategies and potential defenses. Strengthening these models can significantly improve the safety and stability of AI-driven systems in high-stakes environments, promoting greater public trust in AI technologies.

We will make sure that our work will be used ethically and responsibly to lay the foundation for developing robust time series forecasting methods, ultimately contributing to the advancement of secure and reliable AI systems.

REPRODUCIBILITY STATEMENT

We are committed to ensuring the reproducibility of our findings. To this end, we provide a comprehensive description of the proposed Temporally Sparse Attack (TSA), including its formulation, optimization procedure, and evaluation protocols. All experiments are conducted on publicly available datasets that can be accessed through widely used repositories. We also specify the forecasting models used in our evaluation, covering both open-source non-LLM baselines and commercially available LLM APIs.

The implementation of TSA, together with scripts for data preprocessing, evaluation, and visualization, is included in the supplementary material and will be released publicly upon publication to ensure transparency and ease of verification. We additionally report details of the experimental setup (datasets, metrics, baselines), along with extended studies in the appendix to provide deeper insights into the robustness of our conclusions. Collectively, these measures are intended to make our work fully reproducible and to support future research on adversarial robustness in LLM-based time series forecasting.

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

## A    VARIABLES AND DEFINITIONS

In this section, the meaning or definition of each variable is explained in detail in Table 3.

Table 3: Some important variables and their definitions.

| | |
|---|---|
| $d$ | The number of variables |
| $T$ | The length of historical input |
| $L$ | The length of future time series |
| $\tau$ | The number of poisoned time steps for TSA |
| $\mathbf{x}_t$ | $d$-dimentional observations at time $t$ |
| $\mathbf{X}_t$ | A historical time series composed of $T$ observations |
| $\mathbf{Y}_t$ | A time series composed of observations in the next $L$ time steps |
| $\hat{\mathbf{Y}}_t$ | The prediction of future $L$ time steps |
| $f(\cdot)$ | The forecasting model |
| $\rho$ | The adversarial perturbation applied the clean historical time series |
| $\boldsymbol{w}$ | The multiplicative perturbations |
| $\epsilon$ | The scale constraint of perturbations |
| $\mathcal{L}(\cdot,\cdot)$ | The loss function measuring the discrepancy between clean and poisoned prediction |
| $S$ | The indices of nonzero elements in the perturbation vectors |
| $\mathcal{M}(\cdot,\cdot)$ | The merge operator |

## B    EXPERIMENT SETUP

We evaluate the effectiveness of TSA on LLM-based forecasting models across multiple real-world datasets. The experimental design involves three key steps: (i) applying TSA in a manner that preserves the global structure of the time series while misleading the forecasts, (ii) introducing Gaussian White Noise (GWN) as a baseline, where random noise sampled from a normal distribution is added to the input sequence, and (iii) measuring forecasting accuracy using Mean Absolute Error (MAE) and Mean Squared Error (MSE) to capture the extent of performance degradation. All experiments are implemented in PyTorch 1.7.1 with Python 3.7.4, and executed on an Ubuntu 18.04 LTS system equipped with an NVIDIA Tesla V100 GPU.

### B.1    TARGET MODELS

Three representative LLM-based forecasting models, along with three non-LLM-based forecasting models, are included in the experiment to assess the effectiveness of TSA:

- **TimeGPT** (Garza & Mergenthaler-Canseco, 2023): A pre-trained LLM specialized for time series forecasting, incorporating advanced attention mechanisms and temporal encoding to capture complex patterns.
- **LLMTime** (Gruver et al., 2024): A general-purpose LLM adapted for time series forecasting by framing it as a next-token prediction task. We evaluate multiple versions, including those based on GPT-3.5, GPT-4, LLaMA, and Mistral.
- **TimeLLM** (Jin et al., 2024): A model that reprograms time series data into textual inputs for LLMs, leveraging the Prompt-as-Prefix (PaP) technique to enhance forecasting accuracy.
- **TimesNet** (Wu et al., 2023), **TimeMixer** (Wang et al., 2024a), and **TimeXer** (Wang et al., 2024b): non-LLM transformer-based forecasting models introduced to explore the potential impact of our attack on non-LLM models.

These models represent three key strategies for time series forecasting: (1) domain-specific pre-training tailored for time series data (TimeGPT), (2) adapting general-purpose LLMs to forecasting tasks (LLMTime), and (3) input reprogramming to enhance compatibility with LLMs (TimeLLM). Additionally, the inclusion of non-LLM models (TimesNet, TimeMixer, and TimeXer) provides a broader framework for evaluating adversarial robustness across both LLM-based and non-LLM models.

## B.2 BASELINE ATTACKS

**Gaussian White Noise (GWN).** As a simple reference point, we consider injecting noise drawn from a Gaussian distribution directly into the input series. This baseline helps distinguish the impact of unstructured, random perturbations from the targeted and systematic disruptions introduced by TSA.

**Directional Gradient Approximation (DGA).** Following (Liu et al., 2025), DGA is employed as a query-based adversarial method. It estimates gradient information through repeated interactions with the forecasting model and leverages these estimates to craft perturbations. In contrast, TSA requires no access to the target model, highlighting the practicality of a query-free approach that manipulates the tokenization stage instead of relying on gradient exploration.

**Sparse DGA with random position (DGA$_{random}$)** and **Sparse DGA with greedy search (DGA$_{greedy}$).** We construct two sparse variants of DGA. Each variant perturbs the same number of time steps as the proposed TSA, but selects the perturbation positions either through random sampling or via a greedy search strategy (Dong et al., 2020). The perturbation updates follow the same procedure as in the original DGA.

## B.3 DATASETS

Our evaluation makes use of five publicly available datasets, each reflecting unique forecasting challenges across different domains.

The **ETTh1** and **ETTh2** datasets (Zhou et al., 2021) record hourly temperature and electricity load measurements from transformer stations over two years. These series encompass both rapid variations and recurring seasonal patterns, offering a comprehensive benchmark for assessing model accuracy on energy-related forecasting tasks.

The **Traffic** dataset (Gruver et al., 2024) reports hourly vehicle flow counts from the city of Istanbul. Its strong dependence on rush-hour cycles and road usage patterns, combined with high volatility, makes it a demanding test case for time series models.

The **Weather** dataset (Zeng et al., 2023) provides hourly atmospheric readings such as temperature, humidity, and wind. Forecasting here is challenging due to the nonlinear dynamics of meteorological systems, requiring models to account for both short-lived variations and broader climatic tendencies.

The **Exchange Rates** dataset (Lai et al., 2018) covers daily foreign currency exchange values for eight countries between 1990 and 2016. It reflects complex dependencies in global financial markets and is widely used to evaluate long-horizon economic forecasting.

The **Solar** dataset (Lai et al., 2018) consists of solar power output measurements collected in 2006 from 137 photovoltaic plants in Alabama, sampled every 10 minutes. It highlights the fine-scale variability of renewable energy production and the influence of environmental and weather conditions on short-term generation.

Table 4: Detailed dataset descriptions.

| Dataset | Dim | Frequency | Size | Information |
|---------|-----|-----------|------|-------------|
| ETTh1 | 7 | Hourly | 14307 | Electricity |
| ETTh2 | 7 | Hourly | 14307 | Electricity |
| Traffic | 1 | Hourly | 5310 | Transportation |
| Weather | 21 | 10 minute | 52603 | Geoscience |
| Exchange | 8 | Daily | 7207 | Economy |
| Solar | 137 | Hourly | 52179 | Energy |

Across all datasets, we follow a uniform partitioning strategy, using 60% of the samples for training, 20% for validation, and the remaining 20% for testing. Each forecasting model operates under the same setup, where a 96-length input sequence is provided to predict the subsequent 48 time steps, guaranteeing comparability across experiments.

### B.4 METRICS

To assess both forecasting accuracy and the impact of adversarial perturbations, we report results using Mean Absolute Error (MAE) and Mean Squared Error (MSE). Let $\mathbf{Y}_t$ denote the observed value at time step $t$ and $\hat{\mathbf{Y}}_t$ the model's prediction. The two metrics are computed as follows:

$$\text{MAE} = \frac{1}{T} \sum_{t=1}^{T} \left| \hat{\mathbf{Y}}_t - \mathbf{Y}_t \right|, \tag{9}$$

$$\text{MSE} = \frac{1}{T} \sum_{t=1}^{T} \left( \hat{\mathbf{Y}}_t - \mathbf{Y}_t \right)^2, \tag{10}$$

where $T$ is the number of prediction steps. MAE captures the average magnitude of errors in absolute terms, while MSE penalizes larger deviations more heavily by squaring them.

## C EFFECTIVENESS EVALUATION ON DYNAMIC INPUT/OUTPUT LENGTH

This section presents results under a long-sequence setting with prediction lengths ranging from 48 to 1024 on the ETTh1 dataset. MAE is used as the evaluation metric. We include a historical average baseline and a non-LLM-based model (TimesNet) for comparison. Table 5 records the experiment result.

Table 5: Forecasting performance on ETTh1 dataset under different horizons, with and without TSA.

| Models | ETTh1 48 | ETTh1 168 | ETTh1 336 | ETTh1 720 | ETTh1 1024 |
|---|---|---|---|---|---|
| Historical Average | 0.205 | 0.218 | 0.238 | 0.262 | 0.288 |
| TimeGPT (w/o TSA) | 0.192 | 0.334 | 0.391 | 0.474 | 0.497 |
| TimeGPT (w/ TSA) | 0.203 | 0.361 | 0.408 | 0.509 | 0.533 |
| TimesNet (w/o TSA) | 0.202 | 0.346 | 0.375 | 0.496 | 0.512 |
| TimesNet (w/ TSA) | 0.206 | 0.368 | 0.390 | 0.525 | 0.558 |

Our experimental results highlight several important observations. First, the proposed TSA consistently degrades the performance of both LLM-based and transformer-based forecasting models, demonstrating its robustness even under very long prediction horizons. Second, we find that as the forecasting horizon increases, both model families experience substantial error accumulation, in many cases performing worse than the historical average baseline. This outcome echoes concerns raised in prior work about the practicality of extremely long-term forecasting tasks. To ensure fairness and avoid potential controversy, we therefore report our main results under a standardized and widely accepted setting, using an input length of 96 and an output length of 48.

To further examine the robustness of the proposed attack, we evaluate its effectiveness under varying input lengths. In this experiment, LLMTime with GPT-4 is used as the target model, with input windows of 96, 128, 256, 512, and 1024 steps, while the forecasting horizon is fixed at 48. The proposed TSA, which perturbs only 10% of the input sequence, is compared against GWN and DGA, both of which manipulate the full input.

Table 6: Forecasting performance of LLMTime (GPT-4) on ETTh1 under varying input lengths (prediction horizon fixed at 48).

| Models | ETTh1 96/48 | ETTh1 128/48 | ETTh1 256/48 | ETTh1 512/48 | ETTh1 1024/48 |
|---|---|---|---|---|---|
| LLMTime (w/o attack) | 0.202 | 0.201 | 0.197 | 0.211 | 0.205 |
| LLMTime (w/ TSA) | 0.230 | 0.238 | 0.244 | 0.249 | 0.251 |
| LLMTime (w/ GWN) | 0.213 | 0.202 | 0.205 | 0.214 | 0.199 |
| LLMTime (w/ DGA) | 0.232 | 0.241 | 0.249 | 0.252 | 0.258 |

The results in Table 6 reveal several key insights. First, TSA consistently degrades the model's performance across all input lengths, while GWN has only marginal impact. Second, increasing the input length yields only limited accuracy improvements for the clean model, yet provides more opportunities for adversarial methods to introduce harmful perturbations. Finally, despite perturbing far fewer time steps, TSA achieves attack effectiveness comparable to DGA, demonstrating that sparse, structured perturbations are sufficient to substantially degrade forecasting accuracy.

## D EFFECTIVENESS EVALUATION ON VARIATE-WISE FORECASTING

To further substantiate our results, we provide a variate-wise analysis of forecasting performance on the Weather dataset, extending Table 2 from the main submission. In this evaluation, we compare model predictions under clean conditions and under two types of adversarial perturbations: Gaussian White Noise (GWN) and the proposed Temporally Sparse Attack (TSA).

We examine two representative LLM-based forecasters:

- **LLMTime**, implemented with GPT-3.5 as its backbone.
- **TimeGPT**, a pre-trained commercial LLM-based forecasting system.

In the reported results, "+ GWN" refers to forecasts under GWN injection, while "+ TSA" denotes forecasts under our proposed attack. Model performance is quantified using MAE. As summarized in Table 7, TSA consistently leads to a marked increase in prediction error across all variates, underscoring its effectiveness compared to random noise.

Table 7: Comparison of forecasting errors across different variates under clean input, Gaussian White Noise (GWN), and Temporally Sparse Attack (TSA) for LLMTime and TimeGPT.

| Model | LLMTime | | | TimeGPT | | |
|---|---|---|---|---|---|---|
| **Variate** | - | + GWN | + TSA | - | + GWN | + TSA |
| T (degC) | 0.0150 | 0.0152 | **0.0164** | 0.0142 | 0.0147 | **0.0160** |
| Tpot (K) | 0.0162 | 0.0167 | **0.0205** | 0.0153 | 0.0154 | **0.0173** |
| rh (%) | 0.0221 | 0.0227 | **0.0268** | 0.0218 | 0.0221 | **0.0254** |
| VPact (mbar) | 0.0207 | 0.0207 | **0.0221** | 0.0205 | 0.0208 | **0.0220** |
| H2OC (mmol/mol) | 0.0264 | 0.0262 | **0.0311** | 0.0247 | 0.0253 | **0.0304** |
| rho (g/m³) | 0.0176 | 0.0180 | **0.0202** | 0.0160 | 0.0162 | **0.0189** |
| max. wv (m/s) | 0.0008 | 0.0008 | **0.0009** | 0.0007 | 0.0008 | **0.0008** |
| wd (deg) | 0.1022 | 0.1052 | **0.1304** | 0.0986 | 0.0994 | **0.1042** |
| raining (s) | 0.0601 | 0.0598 | **0.0674** | 0.0582 | 0.0590 | **0.0657** |
| SWDR (W/m²) | 0.0177 | 0.0180 | **0.0206** | 0.0173 | 0.0172 | **0.0208** |
| PAR (umol/m²/s) | 0.0351 | 0.0378 | **0.0421** | 0.0324 | 0.0328 | **0.0372** |
| Tlog (degC) | 0.0104 | 0.0121 | **0.0143** | 0.0102 | 0.0108 | **0.0127** |

To avoid further misunderstanding, we want to highlight the distinction between univariate forecasting methods and multivariate forecasting tasks. Although all forecasting models (e.g., LLMTime, TimesNet) adopt a univariate forecasting mechanism, they process each variate separately, effectively treating the multivariate task as multiple univariate tasks.

Please note that changing the forecasting mechanism (from univariate to multivariate) would require re-designing or re-training the models, which is not applicable in our adversarial attack setting. This constraint is especially relevant for commercial LLM-based forecasters like TimeGPT, which only provide API-level access without exposing internal model parameters or allowing architectural modifications.

# E    EFFECTIVENESS–EFFICIENCY TRADE-OFF IN ONE-QUERY AND MULTI-QUERY ATTACKS

Section 4 introduced an SP-based approach for selecting attack positions and generating an FGSM-like (Goodfellow et al., 2015) one-query attack. Although efficient, the one-query attack may be suboptimal in terms of perturbation effectiveness. In contrast, PGD (Madry et al., 2018) enables more flexible and powerful perturbation optimization but requires multiple model queries, making it significantly more costly in black-box settings. This section presents an experiment designed to examine the effectiveness–efficiency trade-off between one-query and multiple-query attacks.

We first provide a PGD-style multi-step extension of the proposed TSA, defined as

$$x^{z+1} = \Pi_{\mathcal{B}_\epsilon(x)}(x^z + \alpha \operatorname{sign}(\hat{g}^z)),\tag{11}$$

where $x^z$ denotes the adversarial example at iteration $z$, and $\hat{g}^z$ represents the surrogate gradient estimated at step $z$ according to Equation 7.

We set the maximum number of iterations to $Z$, and Table 8 summarizes the attack effectiveness and query cost for the one-query attack (proposed) and the multi-query variants with $Z \in [5, 20]$.

Table 8: Balancing effectiveness and efficiency in single-query and multiple-query attacks.

| Datasets/Models | Metrics | clean | one-query | 5-query | 10-query | 15-query | 20-query |
|---|---|---|---|---|---|---|---|
| ETTh1/LLMTime | MAE | 0.202 | 0.230 | 0.230 | 0.228 | **0.232** | 0.231 |
|  | Minute | - | **1.20** | 4.15 | 8.50 | 13.35 | 16.75 |
| Traffic/TimeGPT | MAE | 1.201 | 1.208 | 1.210 | 1.216 | 1.215 | **1.217** |
|  | Minute | - | **0.65** | 2.30 | 4.45 | 6.95 | 8.20 |
| Exchange/TimeLLM | MAE | 0.034 | 0.048 | 0.048 | 0.050 | **0.052** | 0.051 |
|  | Minute | - | **2.35** | 8.60 | 14.85 | 23.70 | 30.25 |

This trade-off experiment shows that the PGD-like multi-step attack can improve attack effectiveness by approximately 5%, but at the expense of incurring more than $11\times$ additional query cost. Therefore, we adopt the one-step attack as a practical compromise between effectiveness and efficiency.

# F    ADDITIONAL SPARSE ATTACK BASELINES

We propose an SP-based method to identify sparse attack positions that optimize adversarial performance. In this section, we compare the proposed approach with two alternative sparse attack baselines, PGD_random and PGD_greedy, which maintain the same number of perturbed steps but determine the perturbation positions through either random selection or a greedy strategy.

Table 9: Comparison of adversarial attack effectiveness between TSA and sparse DGA variants.

| Models | LLMTime w/ GPT-3.5 | | LLMTime w/ GPT-4 | | LLMTime w/ LLaMa 2 | | LLMTime w/ Mistral | | TimeLLM w/ GPT-2 | | TimeGPT (2024) | | TimesNet (non-LLM) | |
|---|---|---|---|---|---|---|---|---|---|---|---|---|---|---|
| Metrics | MSE | MAE | MSE | MAE | MSE | MAE | MSE | MAE | MSE | MAE | MSE | MAE | MSE | MAE |
| Traffic | 0.837 | 0.844 | 0.805 | 0.779 | 0.891 | 1.005 | 0.826 | 0.973 | 0.995 | 1.013 | 1.890 | 1.201 | 1.095 | 1.022 |
| w/ DGA_random | 0.848 | 0.867 | 0.812 | 0.801 | 0.905 | 1.010 | 1.012 | 1.008 | 1.055 | 1.042 | 1.897 | 1.202 | 1.098 | 1.026 |
| w/ DGA_greedy | 0.861 | 0.905 | 0.865 | 8.337 | 0.923 | 1.025 | 1.138 | 1.085 | 1.072 | 1.115 | 1.904 | 1.206 | 1.117 | 1.035 |
| w/ TSA | **0.901** | **1.037** | **1.179** | **1.008** | **0.969** | **1.085** | **1.493** | **1.204** | **1.147** | **1.332** | **1.920** | **1.208** | **1.136** | **1.093** |
| ETTh1 | 0.073 | 0.213 | 0.071 | 0.202 | 0.086 | 0.244 | 0.097 | 0.274 | 0.089 | 0.202 | 0.059 | 0.192 | 0.073 | 0.202 |
| w/ DGA_random | 0.078 | 0.221 | 0.076 | 0.212 | 0.086 | 0.246 | 0.097 | 0.280 | **0.092** | 0.227 | 0.060 | 0.197 | 0.078 | 0.205 |
| w/ DGA_greedy | 0.080 | 0.226 | 0.078 | 0.219 | 0.089 | 0.247 | **0.099** | 0.293 | **0.092** | 0.235 | 0.060 | 0.199 | 0.077 | 0.202 |
| w/ TSA | **0.082** | **0.235** | **0.079** | **0.230** | **0.092** | **0.249** | 0.097 | **0.295** | 0.091 | **0.237** | **0.061** | **0.203** | **0.080** | **0.206** |
| Weather | 0.005 | 0.051 | 0.004 | 0.048 | 0.008 | 0.072 | 0.006 | 0.057 | 0.004 | 0.034 | 0.004 | 0.043 | 0.003 | 0.042 |
| w/ DGA_random | 0.005 | 0.055 | 0.004 | 0.050 | 0.008 | 0.073 | 0.006 | 0.060 | 0.004 | 0.036 | 0.004 | 0.050 | 0.003 | **0.043** |
| w/ DGA_greedy | 0.005 | 0.058 | 0.005 | 0.054 | 0.009 | 0.074 | 0.006 | 0.060 | 0.004 | 0.040 | 0.005 | 0.058 | 0.003 | 0.042 |
| w/ TSA | **0.006** | **0.060** | **0.006** | **0.058** | **0.010** | **0.076** | 0.006 | **0.065** | 0.004 | **0.048** | **0.007** | **0.072** | **0.004** | **0.043** |

**Sparse DGA with Random Position (DGA_random)** and **Sparse DGA with Greedy Search (DGA_greedy).** We design two sparse variants of DGA. Each variant perturbs the same number of time steps as the proposed TSA, but selects the attack positions either through random sampling or

via a greedy search strategy (Dong et al., 2020). The perturbation update rule remains identical to that of the original DGA.

The experimental setup follows Section 5.1. This comparison against plausible sparse baselines further demonstrates the strength of the proposed TSA. As shown in Table 9, TSA consistently outperforms both sparse DGA variants across most settings. Notably, for LLM-based forecasting models, TSA yields approximately 84% larger prediction errors compared with the greedy sparse DGA.

## G   VULNERABILITY COMPARISON BETWEEN LLM-BASED AND NON-LLM-BASED FORECASTERS

This section presents a vulnerability comparison between LLM-based and non-LLM-based time series forecasting models under the proposed TSA attack. The experimental setup follows Section 5.1. We evaluate four LLM-based forecasters and three non-LLM-based forecasters across six datasets.

The results, summarized in Table 10, reveal two key findings: **i.** The proposed TSA consistently degrades the performance of both LLM-based and non-LLM-based models; and **ii.** LLM-based forecasting models are generally more vulnerable to adversarial attacks. A similar observation is reported by Liu et al. (2025). These results suggest that, although LLM-based models offer strong zero-shot forecasting capabilities, their reduced robustness warrants careful consideration in real-world applications.

Table 10: Vulnerability comparison between LLM-based and non-LLM-based time series forecasters.

| Models | LLMTime w/ GPT-3.5 | | LLMTime w/ GPT-4 | | TimeLLM w/ GPT-2 | | TimeGPT (2024) | | TimesNet (2023) | | TimeMixer (2024) | | TimeXer (2024) | |
|---|---|---|---|---|---|---|---|---|---|---|---|---|---|---|
| Metrics | MSE | MAE | MSE | MAE | MSE | MAE | MSE | MAE | MSE | MAE | MSE | MAE | MSE | MAE |
| Traffic | 0.837 | 0.844 | 0.805 | 0.779 | 0.995 | 1.013 | 1.890 | 1.201 | 1.095 | 1.022 | 0.902 | 0.913 | 0.877 | 0.904 |
| w/ GWN | 0.882 | 0.908 | 0.883 | 0.864 | 1.123 | 1.221 | 1.848 | 1.204 | 1.103 | 1.035 | 0.913 | 0.932 | 0.890 | 0.921 |
| w/ TSA | **0.901** | **1.037** | **1.179** | **1.008** | **1.147** | **1.332** | **1.920** | **1.208** | **1.136** | **1.093** | **1.017** | **1.135** | **0.963** | **1.125** |
| ETTh1 | 0.073 | 0.213 | 0.071 | 0.202 | 0.089 | 0.202 | 0.059 | 0.192 | 0.073 | 0.202 | 0.062 | 0.198 | 0.069 | 0.195 |
| w/ GWN | 0.077 | 0.219 | 0.076 | 0.213 | **0.102** | 0.231 | 0.059 | 0.193 | 0.074 | 0.202 | 0.065 | 0.200 | 0.070 | 0.196 |
| w/ TSA | **0.082** | **0.235** | **0.079** | **0.230** | 0.091 | **0.237** | **0.061** | **0.203** | **0.080** | **0.206** | **0.068** | **0.201** | **0.072** | **0.201** |
| ETTh2 | 0.263 | 0.372 | 0.155 | 0.267 | 0.238 | 0.361 | 0.161 | 0.297 | 0.166 | 0.316 | 0.163 | 0.294 | 0.160 | 0.292 |
| w/ GWN | 0.263 | 0.342 | 0.175 | 0.303 | 0.235 | 0.355 | 0.160 | 0.301 | 0.166 | 0.316 | 0.165 | 0.295 | 0.164 | 0.296 |
| w/ TSA | **0.271** | **0.402** | **0.195** | **0.319** | **0.299** | **0.440** | **0.168** | **0.307** | **0.167** | **0.319** | **0.166** | **0.299** | **0.168** | **0.301** |
| Weather | 0.005 | 0.051 | 0.004 | 0.048 | 0.004 | 0.034 | 0.004 | 0.043 | 0.003 | 0.042 | 0.003 | 0.038 | 0.004 | 0.040 |
| w/ GWN | 0.005 | 0.053 | 0.005 | 0.051 | 0.004 | 0.033 | 0.004 | 0.043 | 0.003 | 0.042 | 0.003 | 0.041 | 0.004 | 0.040 |
| w/ TSA | **0.006** | **0.060** | **0.006** | **0.058** | **0.004** | **0.048** | **0.007** | **0.072** | **0.004** | **0.043** | **0.004** | **0.051** | **0.005** | **0.048** |
| Exchange | 0.038 | 0.146 | 0.040 | 0.152 | 0.056 | 0.188 | 0.256 | 0.368 | 0.056 | 0.184 | 0.059 | 0.193 | 0.043 | 0.181 |
| w/ GWN | 0.042 | 0.179 | 0.046 | 0.182 | 0.059 | **0.194** | 0.329 | 0.413 | **0.065** | **0.195** | 0.061 | 0.199 | 0.044 | 0.190 |
| w/ TSA | **0.049** | **0.196** | **0.065** | **0.190** | **0.061** | 0.189 | **0.474** | **0.537** | 0.062 | 0.190 | **0.068** | **0.203** | **0.050** | **0.195** |
| Solar | 0.316 | 0.325 | 0.235 | 0.276 | 0.331 | 0.347 | 0.244 | 0.279 | 0.301 | 0.319 | 0.287 | 0.288 | 0.294 | 0.303 |
| w/ GWN | 0.319 | 0.323 | 0.236 | 0.280 | **0.337** | 0.348 | 0.244 | 0.282 | 0.305 | 0.322 | 0.286 | 0.290 | 0.294 | 0.305 |
| w/ TSA | **0.342** | **0.364** | **0.288** | **0.310** | 0.337 | **0.351** | **0.290** | **0.315** | **0.312** | **0.326** | **0.292** | **0.298** | **0.302** | **0.311** |

## H   TRANSFORMATION FOR TARGETED ATTACKS

The proposed TSA is an untargeted, label-free black-box attack, where attack success is evaluated based on how much the perturbation worsens MAE or MSE. In practice, however, a more realistic adversarial objective is to force the forecasting model to output attacker-specified predictions. This section extends TSA to support such targeted attack goals.

First, the proposed TSA is reformulated into a targeted attack version. The original optimization in Equation 3 becomes:

$$\min_{\boldsymbol{w}} \mathcal{L}(f(\mathbf{X}_t(1+\boldsymbol{w})), \mathcal{Y}_t)$$
$$\text{s.t.} \quad \|\boldsymbol{w}\|_0 = \tau, \quad \|w_i\|_1 \leq \epsilon, \quad i \in [t - T + 1, t], \tag{12}$$

where $\mathcal{Y}_t$ is the attacker-chosen target output.

Next, the gradient estimation in Equation 7 is updated as:

$$\hat{g} = \frac{\mathcal{L}(\mathcal{Y} - \mathcal{F}(\mathbf{X}_t, w_j, \Delta)) - \mathcal{L}(\mathcal{Y} - \mathcal{F}(\mathbf{X}_t, w_j, -\Delta))}{2\Delta}, \tag{13}$$

where $\mathcal{Y}$ denotes the attacker-chosen forecast.

A targeted version of Algorithm 1 is obtained by modifying the loss computation in line 5. Equation 8 is replaced with:

$$\boldsymbol{r} := \mathcal{L}(f(\mathbf{X}_t(1 + \mathcal{M}(\boldsymbol{w}_S, w_j))), \mathcal{Y}_t). \tag{14}$$

With these three modifications, TSA becomes a targeted attack. Attack effectiveness is evaluated through success rate rather than degradation of MAE/MSE. The success indicator at time step $t$ is:

$$l_t^S(\hat{\mathbf{Y}}_t, \mathcal{Y}_t) = \begin{cases} 1, & \|\hat{\mathbf{Y}}_t - \mathcal{Y}_t\|_2 \leq \xi, \\ 0, & \|\hat{\mathbf{Y}}_t - \mathcal{Y}_t\|_2 > \xi, \end{cases} \tag{15}$$

where $\xi$ is a predefined boundary. The overall success rate is computed as:

$$l^S = \frac{\sum l_t^S}{L} \times 100\%, \tag{16}$$

where $L$ is the number of examples.

Table 11: Attack effectiveness on targeted TSA.

| Models | LLMTime w/ GPT-3.5 | | LLMTime w/ GPT-4 | | TimeLLM w/ GPT-2 | | TimeGPT (2024) | |
|---|---|---|---|---|---|---|---|---|
| Metrics | MSE | $l^S$ | MSE | $l^S$ | MSE | $l^S$ | MSE | $l^S$ |
| Traffic | 0.837 | - | 0.805 | - | 0.995 | - | 1.890 | - |
| w/ TSA | 0.898 | 11.2% | 1.174 | 17.8% | 1.140 | 13.5% | 1.933 | 10.4% |
| ETTh1 | 0.073 | - | 0.071 | - | 0.089 | - | 0.059 | - |
| w/ TSA | 0.081 | 9.6% | 0.077 | 8.6% | 0.093 | 4.3% | 0.060 | 2.8% |
| ETTh2 | 0.263 | - | 0.155 | - | 0.238 | - | 0.161 | - |
| w/ TSA | 0.268 | 6.3% | 0.193 | 18.4% | 0.287 | 16.6% | 0.165 | 8.4% |
| Weather | 0.005 | - | 0.004 | - | 0.004 | - | 0.004 | - |
| w/ TSA | 0.006 | 12.7% | 0.006 | 14.9% | 0.004 | 17.1% | 0.007 | 22.3% |
| Exchange | 0.038 | - | 0.040 | - | 0.056 | - | 0.256 | - |
| w/ TSA | 0.044 | 8.5% | 0.063 | 18.8% | 0.059 | 9.7% | 0.455 | 31.4% |
| Solar | 0.316 | - | 0.235 | - | 0.331 | - | 0.244 | - |
| w/ TSA | 0.337 | 13.5% | 0.269 | 17.2% | 0.336 | 5.7% | 0.281 | 16.9% |

We evaluate the targeted TSA on four LLM-based forecasters across six datasets. Two metrics are used to assess attack effectiveness: MSE and the success rate ($l^S$). The results, shown in Table 11, indicate that the targeted TSA still induces substantial degradation in forecasting accuracy. The attack success rate ranges from 3% to over 30%, with an average of approximately 17%. These findings also suggest that the vulnerability of LLM-based forecasters varies considerably across datasets.

# I    COST EVALUATION OF THE SP-BASED SOLUTION

This section compares the computational cost of generating adversarial perturbations using the proposed SP-based solution versus a greedy search strategy. The experiment is conducted on four LLM-based time series forecasters across two datasets, with results summarized in Table 12. The findings show that the proposed SP-based algorithm not only produces more effective perturbations but also requires substantially less computation, reducing cost by roughly 80%. These empirical results are consistent with the theoretical analysis in Section 4.1, where the computational complexity of the SP-based method is $\mathcal{O}(T \times \tau)$, compared to the much higher complexity $\mathcal{O}(T^\tau)$ of a standard greedy algorithm.

Table 12: Attack effectiveness and computational cost comparison between SP-based TSA and sparse DGA with greedy search.

| Models | LLMTime w/ GPT-3.5 | | LLMTime w/ GPT-4 | | TimeLLM w/ GPT-2 | | TimeGPT (2024) | |
|---|---|---|---|---|---|---|---|---|
| Metrics | MSE | Minute | MSE | Minute | MSE | Minute | MSE | Minute |
| Traffic | 0.837 | - | 0.805 | - | 0.995 | - | 1.890 | - |
| w/ DGA$_{greedy}$ | 0.861 | 7.55 | 0.865 | 6.48 | 1.072 | 10.06 | 1.904 | 3.80 |
| w/ TSA | **0.901** | **1.35** | **1.179** | **1.24** | **1.147** | **3.88** | **1.920** | **0.65** |
| Solar | 0.316 | - | 0.235 | - | 0.331 | - | 0.244 | - |
| w/ DGA$_{greedy}$ | 0.325 | 9.03 | 0.248 | 6.98 | 0.332 | 13.76 | 0.260 | 4.92 |
| w/ TSA | **0.337** | **1.90** | **0.269** | **1.41** | **0.336** | **4.25** | **0.281** | **6.28** |

## J UNCERTAINTY ANALYSIS

This section evaluates the uncertainty of the attack by reporting the standard deviation of the increased errors across multiple runs, thereby assessing the reliability of the observed performance gaps.

We evaluate four LLM-based forecasters across three datasets and run the proposed attack 20 times. The results, summarized in Table 13, show that the proposed TSA exhibits strong stability in attack effectiveness. Even at the lower bound of its performance range, TSA consistently induces a substantial degradation in forecasting accuracy.

Table 13: Reliability analysis on attack performance.

| Models | LLMTime w/ GPT-3.5 | | LLMTime w/ GPT-4 | | TimeLLM w/ GPT-2 | | TimeGPT (2024) | |
|---|---|---|---|---|---|---|---|---|
| Metrics | MSE | Variance | MSE | Variance | MSE | Variance | MSE | Variance |
| Traffic | 0.837 | - | 0.805 | - | 0.995 | - | 1.890 | - |
| w/ TSA | 0.892 | ±0.012 | 1.171 | ±0.006 | 1.140 | ±0.010 | 1.914 | ±0.008 |
| ETTh1 | 0.073 | - | 0.071 | - | 0.089 | - | 0.059 | - |
| w/ TSA | 0.081 | ±0.001 | 0.07 | ±0.002 | 0.090 | ±0.002 | 0.061 | ±0.001 |
| Solar | 0.316 | - | 0.235 | - | 0.331 | - | 0.244 | - |
| w/ TSA | 0.334 | ±0.005 | 0.260 | ±0.011 | 0.332 | ±0.007 | 0.269 | ±0.014 |

## K LLM USAGE STATEMENT

We employed ChatGPT-5 solely for language polishing, such as refining grammar and improving readability. At no stage were LLMs used to generate research ideas, construct the attack methodology, carry out experiments, or perform literature review.

All technical elements of this study, including the design and definition of the Temporally Sparse Attack (TSA), the development of algorithms, the setup and execution of experiments, and the subsequent analysis and interpretation, are entirely the authors' own work.

In this research, Large Language Models (LLMs) based time series forecasting models appear only as the *objects of study*, functioning as the forecasting systems that our attack targets. Their role as experimental subjects is fully detailed in Section 5.

