# OpenReview forum: "Temporally Sparse Attack against Large Language Models in Time Series Forecasting"
_ICLR.cc/2026/Conference — Submitted to ICLR 2026_

### Official Review · Reviewer_eCgk · 2025-10-28

**Soundness:** 3
**Presentation:** 2
**Contribution:** 3
**Rating:** 4
**Confidence:** 3

**Summary:**

This paper proposes a Temporally Sparse Attack (TSA) framework that targets large language model (LLM)–based time series forecasters. The central idea is to constrain adversarial perturbations to a limited subset of time steps, formulating the problem as a Cardinality-Constrained Optimization Problem (CCOP) and solving it via an adapted Subspace Pursuit (SP) algorithm with gradient-free optimization. Experiments on six datasets and multiple LLM-based forecasting systems (LLMTime, TimeGPT, TimeLLM, etc.) suggest that perturbing only 10% of the input sequence can substantially degrade forecasting accuracy, even when filter-based defenses are applied. To summarize, the paper is well-motivated in terms of addressing practical attack constraints (sparse, black-box, label-free).

**Strengths:**

[1] The paper focuses on a realistic attack setting—perturbing only a small portion of the time series, which mirrors constraints in real-world scenarios such as financial or sensor systems where continuous manipulation is infeasible. This sparsity assumption makes the problem more relevant than fully dense adversarial perturbations.
[2] The evaluation spans multiple forecasting domains (energy, weather, traffic, economics, and ETTh/ETTm benchmarks), suggesting that the attack framework is not dataset-specific.
[3] The paper clearly states the objective as a cardinality-constrained optimization problem (Equation 2) and emphasizes the trade-off between attack strength and perturbation sparsity. The formalization is neat and readable.
[4] The section discussing defense bypass (filter-based detection and smoothness-based defenses) provides valuable empirical insight, even though the results are preliminary. This helps highlight the vulnerability of time-aware LLMs.

**Weaknesses:**

[1] SP is borrowed from compressive sensing without rigorous reasoning for its advantage in this context. No ablation against simpler greedy or random-search baselines is reported.
[2] The method primarily extends prior black-box attacks by adding a sparsity constraint. While practically interesting, the conceptual contribution is limited compared with existing literature.
[3] The paper depends on commercial APIs (GPT-4, TimeGPT) without specifying prompt templates, token limits, or evaluation protocols. This makes results difficult to reproduce.
[4] Reported MAE/MSE results lack variance or confidence intervals. Given that TSA involves random query sampling, standard deviation across multiple runs is necessary to validate the reliability of performance gaps.
[5] The paper states that TSA is more efficient than greedy methods, but both have complexity. No runtime or query-efficiency comparison is provided.
[6] TSA mainly adds sparsity constraints on top of existing gradient-free attacks. A more explicit comparison to previous black-box methods would help clarify what is fundamentally new.

**Questions:**

[1] Does the finite-difference step require direct access to model outputs? How would this work with commercial APIs that do not expose logits or losses?
[2] Since the paper relies on commercial APIs (TimeGPT, GPT-4), can the authors specify the exact prompting template, temperature, and query limit used during evaluation?
[3] Are the same time steps attacked across models, or does TSA adapt per model architecture?
[4] What is the precise form of the loss used in the black-box query? If true Y_t is unavailable, how is the optimization target defined?

---

> ### Author Response · Authors · 2025-12-03
>
> Thanks so much for your time and insightful comments.
>
> ---
>
> ### **1. Prompt templates, token limits, or evaluation protocols on commercial APIs (weakness 3, question 2)**
>
> We appreciate the opportunity to clarify this. Our work does **not** require designing prompt templates or handling token limits because existing LLM-based forecasting frameworks [1–4] provide well-wrapped APIs for zero-shot forecasting. These systems are used similarly to conventional time series forecasting libraries.
>
> For example, TimeGPT can be queried directly:
>
> ```
> from nixtla import NixtlaClient
> import pandas as pd
>
> nixtla_client = NixtlaClient(api_key='my_api_key')
> df = pd.pd.read_csv() # data
>
> prediction = nixtla_client.forecast(
>     df=df, # data
>     h=12, # prediction horizon
>     freq='MS', # data frequency
>     time_col='timestamp', # copy the time index
>     target_col='value' # column of prediction
> )
> ```
>
> Our study mirrors existing LLM-based forecasting usages, and the code was provided in supply for reproduction.
>
> ---
>
> ### **2. How to compute Loss (questions 1, 4)**
>
> LLM-based forecasting APIs only return predictions. Loss must therefore be computed **locally** without ground truth, consistent with the real-world forecasting workflow.
>
> Because labels at future timesteps are unavailable at runtime, we adopt widely used surrogate labels, such as the Model's own forecast, $Y_{t} \leftarrow f(X_{t})$ (Equation (3) of the manuscript).
>
> We compute the loss using the 1-norm: $\|f(X^*_{t}) - f(X_{t}) \|_{1}$.
>
> Our goal is to **maximize the discrepancy** between predictions from clean and perturbed inputs, under two constraints:
> 1. Poisoned input must remain close to the original input;
> 2. Perturbations must be temporally sparse.
>
> ---
>
> ### **3. Are the same time steps attacked across models? (question 3)**
>
> No. The attacked time steps vary across Models, Datasets, and Input windows.
>
> In the original submission, the left side of Figure 2 shows two examples with distinct perturbation patterns. The bottom of Figure 4 provides a statistical analysis, revealing that many attacks concentrate in the most recent 25% of time steps.
>
> ---
>
> ### **4. Ablation and computational cost studies on SP (weakness 1, 5)**
>
> We sincerely appreciate this insightful suggestion. To address these points, we added:
>
> #### **New Sparse Baselines**
> 1. Sparse DGA (random): Same number of perturbed steps as TSA, but positions chosen randomly.
> 2. Sparse DGA (greedy): Time steps chosen via a greedy heuristic.
>
> #### **Results**
> In **Appendix F**, we compare TSA against these baselines across seven models and three datasets. TSA achieves:
> - 84% larger prediction errors than greedy sparse DGA,
> - 127% larger errors than random sparse DGA,
>
> demonstrating its substantially superior sparse attack capability.
>
> #### **Computational Cost**
> **Appendix I** compares the proposed SP-based method with the greedy strategy. Results show:
> - SP generates more effective perturbations, and
> - Requires **~80% less computation**.
>
> These findings validate both the effectiveness and efficiency of the SP-based design.
>
> ---
>
> ### **5. Reported MAE/MSE results lack variance or confidence intervals (weakness 4)**
>
> We agree that reporting uncertainty is important, not only for our work but for the forecasting and adversarial learning communities more broadly.
>
> In **Appendix J**, we added an uncertainty analysis by reporting the standard deviation of increased errors across 20 runs.
> Results show that even at the performance lower bound, TSA consistently induces substantial degradation in forecasting accuracy.
>
> ---
>
> ### **6. Conceptual contribution relative to existing adversarial attacks (weakness 2, 6)**
>
> We respectfully clarify two key distinctions:
>
> #### **1. Black-box CV/NLP attacks are not directly applicable**
> Most rely on:
> - Label access,
> - Model structure (transfer attack),
> - Known training data.
>
> In LLM-based time series forecasting:
> - Labels are unavailable at runtime,
> - Model structure and weights are inaccessible,
> - Training data cannot be reconstructed.
>
> #### **2. Adding sparsity makes the problem fundamentally different**
> Introducing a temporal sparsity constraint transforms the attack from a standard convex formulation into a **non-convex, NP-hard** optimization problem.
>
> While the mathematical changes may appear small, the resulting attack pipeline and optimization landscape are fundamentally different.
>
> This is why our work relies on a Subspace Pursuit–based strategy tailored to these constraints.
>
> ---
>
> ## **References**
>
> [1] *Time-LLM: Time Series Forecasting by Reprogramming Large Language Models*, ICLR 2024
> [2] *FSTLLM: Spatio-Temporal LLM for Few-Shot Time Series Forecasting*, ICML 2025
> [3] *Chronos: Learning the Language of Time Series*, TMLR 2024
> [4] *Large Language Models Are Zero-Shot Time Series Forecasters*, NeurIPS 2023

---

### Official Review · Reviewer_AhJB · 2025-10-31

**Soundness:** 3
**Presentation:** 3
**Contribution:** 2
**Rating:** 4
**Confidence:** 4

**Summary:**

This paper proposes a Temporally Sparse Attack (TSA) targeting black-box, label-free LLM-based time series forecasting models. TSA formulates the attack as a cardinality-constrained optimization problem and solves it using an adapted Subspace Pursuit algorithm with gradient-free optimization. Experiments on six datasets and several LLM-based forecasters (e.g., TimeGPT, TimeLLM, LLMTime) show that perturbing a few input time steps can significantly degrade forecasting accuracy. While the idea is interesting and the formulation is clear, I have concerns about the paper’s generality, evaluation fairness, and the real-world relevance of the proposed threat model.

**Strengths:**

1. The attack formulation is mathematically sound, and the adapted SP-based algorithm is conceptually neat.
2. Experiments are broad in scope (multiple datasets and LLMs) and demonstrate consistent degradation in performance.
3. Writing is clear, and figures are helpful in conveying the main idea.

**Weaknesses:**

1. The proposed method seems designed specifically for black-box LLM-based forecasters. However, in practical applications, non-LLM models still dominate time series forecasting in both research and industry. TSA’s formulation (black-box, gradient-free, label-free) appears general enough to apply to any model that outputs predictions, not just LLMs.
2. While including TimesNet as a non-LLM baseline is appreciated, TimesNet is now 3 years old. The Time-Series-Library built from TimesNet already includes many more modern models. Testing TSA on a few of these newer non-LLM forecasters would help demonstrate its generality and practical relevance. This should be feasible, as these implementations are readily available in that package. Without such comparisons, it’s hard to tell whether TSA exposes vulnerabilities unique to LLMs or simply general weaknesses shared by all forecasters.
3. The comparison between TSA and DGA (full-series) attacks seems unbalanced. It is not surprising that a full-series attack achieves larger performance degradation. A fairer test would be to sparsify DGA, for example, keeping only the same number of perturbed steps (e.g., 9 time steps) either at random or aligned with TSA’s positions, and then compare the results. This would show whether TSA’s advantage comes from better perturbation selection or simply from fewer overall changes.
4. The paper measures attack success by how much it worsens MAE/MSE, but this doesn’t necessarily reflect a realistic attack objective. In practice, such random degradation is easily detectable and rarely beneficial to attackers. Realistic adversarial objectives in time series forecasting often involve targeted manipulation (e.g., introducing trigger patterns that make the model output attacker-chosen forecasts such as “predicting stock increase” when a certain trigger appears). I would like to see a discussion of how TSA might extend to or inspire such trigger-based or targeted attacks, which are more aligned with real-world threats.

**Questions:**

Please my suggestions in weaknesses.

---

> ### Author Response · Authors · 2025-12-03
>
> Thanks so much for your time and insightful comments. We address each concern in detail below.
>
> ---
>
> ### **1. LLM-specific**
>
> You're right that non-LLM forecasters make up the majority in both academia and industry, and the proposed TSA can manipulate both LLM-based and non-LLM-based forecasters. However, the **LLM-specific focus is intentional and meaningful**:
>
> 1. LLM-based time series forecasting is a rapidly growing research area, yet its adversarial vulnerabilities remain underexplored.
>
> 2. TSA is particularly essential in the LLM setting:
>    - Non-LLM forecasters often have public architectures and weights, enabling white-box or transfer-based black-box attacks.
>    - In contrast, LLM-based forecasters are typically proprietary, black-box systems, making label-free query attacks critical for practical threat modeling.
>    - The training data for LLM-based forecasters are typically large-scale and unavailable, whereas non-LLM forecasters rely on small, domain-specific datasets.
>    - In LLM-specific settings, attackers cannot access these training data, which introduces a new challenge for adversarial attack design.
>
> Based on these motivations, our original manuscript primarily focused on LLM-based forecasters, while including TimesNet as a representative non-LLM baseline. We fully agree that an extended study covering more non-LLM models is valuable.
>
> ---
>
> ### **2. More modern non-LLM baselines than TimesNet(2023)**
>
> We greatly appreciate this suggestion. We have incorporated Timemixer [8] and TimeXer [9] as additional non-LLM baselines.
>
> In **Appendix G**, we now compare four LLM-based and three non-LLM-based models across multiple datasets. The results show that, although LLM-based models exhibit strong zero-shot forecasting performance, they demonstrate reduced adversarial robustness, highlighting an important vulnerability for deployment.
>
> ---
>
> ### **3. Unbalanced comparison between TSA and full-series attack DGA**
>
> We appreciate the reviewer’s insight regarding baselines. In response, we added two sparse DGA variants:
>
> - Sparse DGA (random): Perturbs the same number of time steps as TSA but selects positions randomly.
> - Sparse DGA (greedy): Uses a greedy strategy to determine the most impactful time steps.
>
> In **Appendix F**, we compare TSA against these baselines over seven models and three datasets. TSA achieves:
>
> - 84% larger prediction errors than greedy sparse DGA
> - 127% larger errors than random sparse DGA
>
> These results clearly demonstrate that TSA provides substantially stronger temporally sparse adversarial capabilities.
>
> ---
>
> ### **4. Measurement of attacks and analysis of targeted attacks**
>
> We agree that targeted attacks provide a more realistic evaluation of the attacker’s capabilities. While prior work often measures attack effectiveness using MSE/MAE, which reflect the standard metrics for forecasting accuracy, we appreciate the reviewer’s point about the real-world implications of targeted attacks.
>
> In **Appendix H**, we introduce a targeted version of TSA and evaluate it across four LLM-based forecasters and six applications. Results show:
>
> - Targeted TSA still causes substantial degradation in forecasting accuracy.
> - The average attack success rate reaches ~17%, indicating that attackers can meaningfully steer the model outputs toward chosen targets.
>
> ---
>
> ### **References**
>
> [1] *Time-LLM: Time Series Forecasting by Reprogramming Large Language Models*, ICLR 2024
> [2] *FSTLLM: Spatio-Temporal LLM for Few-Shot Time Series Forecasting*, ICML 2025
> [3] *Chronos: Learning the Language of Time Series*, TMLR 2024
> [4] *Adversarial Attacks on Probabilistic Autoregressive Forecasting Models*, ICML 2020
> [5] *Practical Adversarial Attacks on Spatiotemporal Traffic Forecasting Models*, NeurIPS 2022
> [6] *Robust Multivariate Time-Series Forecasting: Adversarial Attacks and Defense Mechanisms*, ICLR 2023
> [7] *Adversarial Vulnerabilities in Large Language Models for Time Series Forecasting*, AISTATS 2025
> [8] *Timemixer: Decomposable Multiscale Mixing for Time Series Forecasting*, ICLR 2024
> [9] *TimeXer: Empowering Transformers for Time Series Forecasting with Exogenous Variables*, NeurIPS 2024

---

### Official Review · Reviewer_Xcc1 · 2025-10-31

**Soundness:** 3
**Presentation:** 3
**Contribution:** 2
**Rating:** 4
**Confidence:** 3

**Summary:**

This paper proposes a Temporally Sparse Attack against LLM-based time series forecasting models in a black-box manner without training data. TSA is modeled by $l_0$-norm constraint, and a black-box manner without training data is learned by gradients estimation with random Gaussian noise.

**Strengths:**

1. This paper is easy to follow.
2. Clear experimental settings.
3. Well-structured experiments on six real-world open datasets.

**Weaknesses:**

1. Sparse Attack is well studied in CV and NLP. This work ultilizes these existing techniques, i.e. $l_0$-norm constraint, Subspace Pursuit (SP), black-box gradients estimation and Fast Gradient Sign Method (FGSM), on LLM-based time series forecasting models. The contributions and insights of this paper is limited.

2. There are no experiments under defense methods. There are many adversarial defense methods against Sparse Attack. The authors should evaluate their performance under adversarial defense methods to show how their methods works.

**Questions:**

see weaknesses.

**Details Of Ethics Concerns:**

There may be research integrity issues. Too many overlaps with published workshop paper [1] and no reference to [1].

[1] Temporally Sparse Attack for Fooling Large Language Models in Time Series Forecasting. ICLR 2025 Workshop BuildingTrust.

---

> ### Author Response · Authors · 2025-12-03
>
> Thanks for your time. We first address the Ethics Concern and then respond to the two weaknesses.
>
> ---
>
> ### **1. Ethics Concern**
>
> > *There may be research integrity issues. Too many overlaps with published workshop paper [1] and no reference to [1].*
>
> We confirm that our submission has **no research integrity issues**, including plagiarism or dual submission.
>
> The ICLR 2025 Workshop *BuildingTrust* explicitly states that accepted papers are **Non-Archival** ([link](https://building-trust-in-llms.github.io/iclr-workshop/cfp.html)), which avoids dual submission.
>
> ---
>
> ### **2. Limited Contributions**
>
> > *Sparse attack is well studied in CV and NLP. This work utilizes existing techniques. The contributions and insights are limited.*
>
> We respectfully disagree and clarify the novelty from three perspectives: technical contribution, new challenges, and significance.
>
> #### **2.1 Direct Technical Contribution**
> - To our knowledge, this is the first work to formulate the adversarial attack process as a Cardinality-Constrained Optimization Problem (CCOP).
> - This is the first work to introduce Subspace Pursuit (SP) into adversarial attack design.
>
> These represent direct methodological contributions to adversarial machine learning.
>
> #### **2.2 New Challenges Unique to LLM-Based Time Series Forecasting**
> Although sparse attacks have been studied in CV and NLP, this work imposes four fundamental constraints, two of which have not been considered in those domains:
>
> - **Label-free inference**
>    At runtime, no one can access labels as the ground-truth is the future for forecasting applications; classical attacks relying on label gradients cannot be applied.
> - **No access to model structure or parameters**
>    This is the standard black-box setting.
> - **No access to training data**
>    LLM-based forecasters often use proprietary or unknown training datasets.
> - **Sparse and black-box jointly**
>    These constraints make the attack a non-convex, NP-hard optimization problem, which has rarely been addressed in prior works.
>
> Our proposed method is specifically designed to overcome these new challenges.
>
> #### **2.3 Significance**
> LLM-based time series forecasting is rapidly gaining traction:
>
> - Both NeurIPS 2024 and ICLR 2025 hosted workshops specifically on LLM-based forecasting.
> - Companies such as Amazon and startups like Nixtla (TimeGPT) recently released commercial LLM forecasters.
>
> Despite this momentum, their adversarial vulnerabilities remain largely unknown. Understanding their risks is essential for safe deployment in critical applications such as finance, energy, transportation, and climate forecasting.
>
> ---
>
> ### **3. Lack of Experiments Under Adversarial Defense**
>
> > *There are no experiments under defense methods. The authors should evaluate the defense performance to show how the attack works.*
>
> We respectfully clarify that **Section 5.3 ("Mitigation Bypassing Test")** in the original submission **does** evaluate TSA under multiple adversarial defenses.
>
> - We tested TSA against three filter-based defenses.
> - Results show that while these defenses can mitigate full-series attacks, they fail to recover accuracy under TSA.
> - The reason is that TSA modifies only a small number of time steps, thereby bypassing the statistical assumptions underlying most filtering defenses.
>
> Thus, we have already evaluated and demonstrated the limitations of existing defenses against the proposed TSA.
>
> ---
>
> ### **Reference**
> [1] *Temporally Sparse Attack for Fooling Large Language Models in Time Series Forecasting.* ICLR 2025 Workshop BuildingTrust.

---

### Official Review · Reviewer_Csz4 · 2025-10-31

**Soundness:** 3
**Presentation:** 3
**Contribution:** 2
**Rating:** 4
**Confidence:** 4

**Summary:**

This paper proposes a Temporally Sparse Attack (TSA) that effectively degrades the performance of LLM-based time series forecasters by perturbing only a small subset of input time steps. The attack is formulated as a cardinality-constrained optimization problem and solved via an adapted Subspace Pursuit algorithm under a realistic black-box, label-free setting.

**Strengths:**

- Novel Formulation and Solution: Casting the temporally sparse attack as a CCOP and adapting Subspace Pursuit (originally designed for white-box LASSO) to a black-box adversarial setting is an elegant and non-trivial methodological contribution.
- Strong Adversarial Impact with High Stealthiness: The paper shows that perturbing a quite small part of the time steps can severely degrade forecasting performance, and that TSA is more effective than Gaussian white noise and harder to defend against using standard filtering techniques.
- Interpretability and Visualization: The authors include detailed visualizations that compare input/output distributions, and prediction errors, providing strong intuitive support for their claims.

**Weaknesses:**

1. **Suboptimal perturbation update**: The paper adopts an FGSM-like one-step update, resulting in identical perturbation magnitudes across all attacked time steps. This uniformity may lead to suboptimal solutions. A multi-step approach such as PGD could provide more flexible and effective perturbation optimization.
2. **Limited consideration of temporal interaction**: The current approach perturbs one timestamp at a time, ignoring potential synergistic effects between multiple time steps. Jointly selecting and optimizing perturbations across multiple timestamps, rather than in a sequential manner, may lead to stronger and more coordinated attacks.
3. **Unclear evaluation of attack strength**: The effectiveness of TSA is not clearly demonstrated. In Table 2, TSA consistently outperforms a weak heuristic baseline (GWN) but still underperforms compared to DGA on certain datasets. While this comparison is admittedly unfair due to the dense nature of DGA versus the sparsity constraint in TSA, it still leaves reviewers uncertain about the absolute strength of TSA. A more appropriate evaluation would involve comparisons with sparse adversarial attack methods, such as [1,2].
4. **Lack of LLM-specific design**: The proposed method is tailored for general black-box models and does not incorporate any design elements unique to LLMs. As a reviewer, I find this positioning somewhat confusing: why emphasize LLMs specifically if the method is equally applicable to any black-box forecaster, given the absence of LLM-specific methodological considerations?

[1] Sparse and imperceivable adversarial attacks
[2] GreedyFool: Distortion-Aware Sparse Adversarial Attack

**Questions:**

See the weakness

---

> ### Author Response · Authors · 2025-12-03
>
> We sincerely appreciate the reviewer’s constructive insights. Below, we address each concern in detail.
>
> ---
>
> ### **1. Suboptimal perturbation update**
>
> > *One-step attack might be suboptimal, and a multiple-step attack could be more effective.*
>
> We agree that a PGD-like multi-step attack is typically more effective than an FGSM-like one-step attack, but it also incurs substantially higher query costs. The proposed one-step TSA is intentionally designed as a more query-efficient alternative.
>
> To clarify this trade-off, we added an empirical analysis in **Appendix E**, evaluating the effectiveness–efficiency balance between one-step and multi-step attacks. The results show that:
>
> - A PGD-like multi-step attack improves effectiveness by about **5%**,
> - But requires over **11× more queries**.
>
> Thus, we adopt the one-step TSA as a practical compromise that achieves strong performance under strict black-box query constraints.
>
> ---
>
> ### **2. Temporal interaction consideration**
>
> > *The current approach perturbs one timestamp at a time, ignoring potential synergistic effects between multiple time steps.*
>
> We respectfully clarify that TSA **does generate multi-step perturbations** and explicitly considers temporal interactions. For example:
>
> - In Section 5.1 (e.g., Table 2, Figure 2), we perturb 9 out of 96 time steps.
> - The right panel of Figure 6 analyzes attack effectiveness under sparsity levels from 1 to 12 out of 96.
>
> Temporal coupling arises directly from the Subspace Pursuit-based selection mechanism, which jointly determines the most adversarial combination of time steps. This captures synergistic perturbation patterns via iterative refinement, rather than through qualitative assumptions. To further support understanding, the bottom of Figure 4 in the original submission illustrates the poison distribution across time, revealing meaningful temporal structures.
>
> ---
>
> ### **3. Unclear evaluation of attack strength**
>
> > *A more appropriate evaluation would involve comparisons with sparse adversarial attack methods.*
>
> Your suggestion is constructive. Existing sparse attacks primarily target **white-box, label-available computer vision tasks**, and we cannot directly apply them to black-box, label-free LLM-based forecasting.
>
> To address this concern comprehensively:
>
> - We added a new discussion in Related Work analyzing prior sparse CV attacks [1-2] and explaining why their assumptions (white-box, labeled data) do not transfer to LLM-based forecasting.
> - More importantly, we implemented two sparse variants of DGA that select the attack positions either through random sampling or via a greedy search strategy.
>
> In **Appendix F**, we compare TSA against these baselines on seven models across three datasets. TSA achieves **84% larger prediction errors** than greedy sparse DGA, and **127% larger errors** than random sparse DGA. These results clearly indicate that TSA provides substantially stronger sparse adversarial capabilities.
>
> ---
>
> ### **4. LLM-specific design**
>
> > *The proposed method is tailored for general black-box forecasting models.*
>
> TSA can manipulate both LLM-based and non-LLM-based forecasters. However, the **LLM-specific focus is intentional and meaningful**:
>
> 1. LLM-based time series forecasting is a rapidly growing research area, yet its adversarial vulnerabilities remain underexplored.
>
> 2. TSA is particularly essential in the LLM setting:
>    - Non-LLM forecasters often have public architectures and weights, enabling white-box or transfer-based black-box attacks.
>    - In contrast, LLM-based forecasters are typically proprietary, black-box systems, making label-free query attacks critical for practical threat modeling.
>
> To further support this point, **Appendix G** includes a vulnerability comparison of four LLM-based and three non-LLM-based models. The results show that, although LLM-based models provide strong zero-shot forecasting performance, their reduced adversarial robustness requires careful consideration in deployment.
>
> ---
>
> ## **References**
>
> [1] Sparse and Imperceivable Adversarial Attacks.
> [2] GreedyFool: Distortion-Aware Sparse Adversarial Attack.
> [3] Adversarial Vulnerabilities in Large Language Models for Time Series Forecasting.

---

### Author Response · Authors · 2025-12-03

### **1. Why this submission matters**

**Background**: Large Language Models (LLMs) have recently emerged as strong candidates for foundation models in time series forecasting, largely due to their impressive zero-shot forecasting capabilities.

**Gap**: Despite rapid progress, the vulnerabilities of LLM-based forecasting systems remain underexplored. Particularly, existing adversarial attacks on LLM-based forecasters typically perturb the **entire** input sequence, which reduces stealthiness and limits practicality in real-world deployments.

**Proposed solution**: We introduce a temporally sparse, black-box, label-free adversarial attack that manipulates LLM-based time series forecasters by perturbing only 10% of the input, significantly improving stealthiness while retaining strong attack strength.

**Technical contribution**:
- We formulate the attack as a Cardinality-Constrained Optimization Problem (CCOP), which, to the best of our knowledge, is the first application of this formulation in adversarial attack research.
- We adapt Subspace Pursuit (SP), originally developed for white-box Lasso problems, to the black-box, label-free setting.
Together, CCOP and SP constitute new methodological contributions to adversarial machine learning.

**Empirical study**:
We conduct a comprehensive evaluation across 6 LLM-based models and 3 non-LLM models spanning 6 real-world forecasting applications. Results show that the proposed temporally sparse attack is highly effective and, notably, more capable of bypassing filter-based adversarial defenses.

**Overall contribution**:
This work uncovers a critical and previously unaddressed vulnerability: by perturbing only a small portion of the input time series, an attacker can substantially degrade the performance of LLM-based forecasters. This finding raises important concerns about the robustness of emerging foundation models for time series tasks.

---

### **2. Rebuttal summary**

We sincerely appreciate the reviewers’ recognition of the novelty in both the CCOP formulation and the SP-based solution. The main concerns centered on the need for more extensive experimentation to highlight the strength and efficiency of the proposed TSA. In response, we conducted a series of new experiments and analyses:

- **Appendix E**: Empirical evaluation of the trade-off between effectiveness and efficiency for one-step vs. multi-step attacks.
- **Appendix F**: Comparison with two sparse DGA-based baselines.
- **Appendix G**: Vulnerability comparison between LLM-based and non-LLM-based forecasters.
- **Appendix H**: Extension of TSA to a targeted attack setting to assess whether attackers can force models to output attacker-specified predictions.
- **Appendix I**: Computational cost comparison between the SP-based method and a greedy strategy.
- **Appendix J**: Reliability and variance analysis across 20 repeated attack runs.

Modifications are highlighted in blue in the updated version.

---

### **3. Appreciation statement**

We regret that we were unable to engage in real-time discussion with the reviewers, as we invested considerable time running the suggested experiments, and the discussion period unexpectedly closed due to the ICLR “black swan” event.

Nevertheless, we feel fortunate to have received exceptionally high-quality reviews, which bring our submission to the next level. We sincerely thank you for this thoughtful feedback.

---

### Meta-Review · Area_Chair_xWQF · 2025-12-23

**Summary:**

This paper proposes a Temporally Sparse Attack (TSA) for LLM-based time series forecasting.

Strengths:
(1) novel formulation and proposed solution, (2) strong adversarial impact with high stealthiness, (3) overall clear writing, (4) well structured experimental setup with different datasets and LLMs.

Weaknesses:
(1) suboptimal perturbation, (2) limited temporal interaction, (3) limited technical novelty and insight of sparse attack, (4) lack of defense method.

**Reviewer Concerns:**

Most of the concerns/weaknesses were not adequately addressed.

**Reviewer Scores:**

It is unlikely that reviewers will change their scores.

---

### Decision · Program_Chairs · 2026-01-26

Reject